# Number and relative abundance of synaptic vesicles in functionally distinct priming states determine synaptic strength and short-term plasticity

Kun-Han Lin[1] , Mrinalini Ranjan[2,3] , Noa Lipstein[4] , Nils Brose[2], Erwin Neher[1] and Holger Taschenberger[2]

[1]*Laboratory of Membrane Biophysics, Max Planck Institute for Multidisciplinary Sciences, Göttingen, Germany*
[2]*Department of Molecular Neurobiology, Max Planck Institute for Multidisciplinary Sciences, Göttingen, Germany*
[3]*Göttingen Graduate School for Neurosciences, Biophysics, and Molecular Biosciences, Göttingen, Germany*
[4]*Leibniz-Forschungsinstitut für Molekulare Pharmakologie (FMP), Berlin, Germany*

Handling Editors: Katalin Toth & Samuel Young

The peer review history is available in the Supporting Information section of this article (https://doi.org/10.1113/JP286282#support-information-section).

*The Journal of Physiology*

**Abstract figure legend** Neurotransmitter release was characterized at post-hearing rat calyx of Held synapses, before and after pharmacologically augmenting synaptic strength, by recording unitary EPSCs, or capacitance changes in response to presynaptic step depolarizations. Non-negative tensor factorization (NTF) was used to decompose EPSC-derived release time courses measured during stimulus trains into kinetic components representing contributions of functionally distinct synaptic vesicle subpools. A sequential two-step vesicle priming and fusion scheme (LS–TS model) is capable of reproducing experimentally induced changes in synaptic strength and short-term plasticity with a small number of plausible model parameter changes. The combination of NTF-decomposition analysis and state modelling allows one to separate experimentally induced changes in vesicle priming kinetics from those in vesicle fusion probability. The LS–TS model may, thus, provide a framework for a mechanistic analysis of synaptic plasticity and its modulation at different types of synaptic connections.

The Journal of Physiology

**Abstract** Heterogeneity in synaptic strength and short-term plasticity (STP) was characterized in post-hearing rat calyx of Held synapses at near-physiological external $[Ca^{2+}]$ under control conditions and after experimentally induced synaptic potentiation. Kinetic modelling was combined with non-negative tensor factorization (NTF) to separate changes in synaptic vesicle (SV) priming kinetics from those in SV fusion probability ($p_{fusion}$). Heterogeneous synaptic strength and STP under control conditions can be fully accounted for by assuming a uniform $p_{fusion}$ among calyx synapses yet profound synapse-to-synapse variation in the resting equilibrium of SVs in functionally distinct priming states. Although synaptic potentiation induced by either elevated resting $[Ca^{2+}]_i$, elevated external $[Ca^{2+}]$ or stimulation of the diacylglycerol (DAG) signalling pathway leads to seemingly similar changes, that is, stronger synapses with less facilitation and more pronounced depression, the underlying mechanisms are different. Specifically, synaptic potentiation induced by the DAG mimetic and Munc13/PKC activator phorbol 12,13-dibutyrate (PDBu) only moderately enhances $p_{fusion}$ but strongly increases the abundance of fusion-competent maturely primed SVs, demonstrating that the dynamic equilibrium of differentially primed SVs critically determines synaptic strength and STP. Activation of the DAG pathway not only stimulates priming at resting $[Ca^{2+}]_i$ but further promotes SV pool replenishment at elevated $[Ca^{2+}]_i$ following pool-depleting stimulus trains. A two-step priming and fusion scheme which recapitulates the sequential build-up of the molecular SV fusion machinery is capable of reproducing experimentally induced changes in synaptic strength and STP in numerical simulations with a small number of plausible model parameter changes.

(Received 22 November 2024; accepted after revision 20 February 2025; first published online 22 March 2025)

**Corresponding author** Holger Taschenberger, Department of Molecular Neurobiology, Max Planck Institute for Multidisciplinary Sciences, Göttingen, Germany. Email: taschenberger@mpinat.mpg.de

## Key points

- A relatively simple two-step synaptic vesicle (SV) priming and fusion scheme is capable of reproducing experimentally induced changes in synaptic strength and short-term plasticity with a small number of plausible parameter changes.
- The combination of non-negative tensor factorization (NTF)-decomposition analysis and state modelling allows one to separate experimentally induced changes in SV priming kinetics from those in SV fusion probability.
- A relatively low sensitivity of the SV priming equilibrium to changes in resting $[Ca^{2+}]_i$ suggests that the amplitude of the 'effective' action potential (AP)-induced $Ca^{2+}$ transient is quite large, likely representing contributions of global and local $Ca^{2+}$ signals.
- Enhanced synaptic strength and stronger depression after stimulation of the diacylglycerol (DAG) signalling pathway is primarily caused by enhanced SV priming, leading to increased abundance of maturely primed SVs at rest with comparably small changes in SV fusion probability.
- Application of DAG mimetics enhances the $Ca^{2+}$-dependent acceleration of SV priming causing a faster recovery of synaptic strength after pool-depleting stimuli.

**Kun-Han Lin** is a postdoctoral scientist at the Max Planck Institute for Multidisciplinary Sciences in Göttingen, Germany. His research primarily focuses on synaptic physiology, synaptic vesicle dynamics, short-term plasticity and presynaptic $Ca^{2+}$ signalling at the calyx of Held synapse, employing pre- and postsynaptic patch-clamp recording, presynaptic $Ca^{2+}$ imaging and transgenic mouse models. Dr Lin also studies the function of neurons and cardiomyocytes derived from patient-specific induced pluripotent stem cells. His work, thus, bridges synaptic and cardiac physiology, and stem cell biology, aiming to advance the understanding of synaptic transmission and excitation-contraction coupling in health and disease.

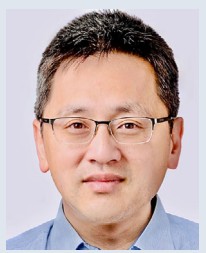

## Introduction

Signalling between nerve cells at chemical synapses involves action potential (AP)-triggered release of neurotransmitter from fusion-competent synaptic vesicles (SVs) at presynaptic active zone (AZ) release sites (Katz, 1969; Südhof, 2014). The molecular processes generating fusion-competent SVs are tightly controlled by multiple proteins that regulate SV docking and priming (Brunger et al., 2018; Rizo, 2022). Evidence from biochemistry, molecular biology, electron microscopy and physiology indicates that the SV priming process represents a reversible transition through several sequential molecular states, in which the final stage of release-machinery maturation is characterized by tight docking of SVs to the AZ membrane (Chang et al., 2018; He et al., 2017; Held et al., 2024; Imig et al., 2014; Kusick et al., 2022; Miki et al., 2018; Papantoniou et al., 2023; Prinslow et al., 2019; Silva et al., 2024; Witkowska et al., 2021; Zenisek et al., 2000). The transition rates between individual priming states may be subject to dynamic regulation by presynaptic cytosolic $Ca^{2+}$ levels ($[Ca^{2+}]_i$) and second messengers to create functionally diverse synapses and to balance supply and consumption of fusion-competent SVs during bouts of synaptic activity (Kusick et al., 2022; Neher & Brose, 2018; Nusser, 2018; Silva et al., 2021).

The molecular complexity of the build-up of the synaptic release machinery indicates that the commonly used model for describing the amount of quantal release (quantal content $m$) in response to single APs as the product of the total number of quanta ($N$) available, which is fixed, and the probability ($p$) for each of these quanta to be released by an AP ($m = N \times p$) represents an oversimplification (Fesce, 1999). Instead, $m$ may be determined by both $p$ and the fraction of vesicles with a mature release machinery among all docked and primed SVs. The stochastic aspects of a reversible molecular transition leading to maturely primed SVs may, at least in part, account for the stochastic properties of quantal release (Fesce, 1999; Scheuss & Neher, 2001; Vere-Jones, 1966).

For example, assuming that $N$ constitutes a dynamic and $Ca^{2+}$-dependent equilibrium of vesicles in two priming states, either tightly (TS) or loosely (LS) docked to the AZ membrane, of which only SVs in state TS (denoted as TS for brevity from here on) represent a mature and fusion-competent state (Neher & Brose, 2018), the probability for AP-triggered SV fusion critically depends on the relative fraction $f_{TS} = TS/(LS + TS)$. In this scenario release probability represents the compound probability of two stochastically independent events occurring together: (1) a docked SV being in the mature, tightly docked priming state at the time of AP arrival ($f_{TS}$) and (2) a tightly docked SV fusing in response to the AP ($p_{fusion}$). Importantly, $f_{TS}$ and $p_{fusion}$, which reflect the outcome of two different molecular processes (vesicle priming *vs.* fusion), may quite possibly be controlled by non-overlapping sets of regulatory proteins. How synapses make use of the regulation of $f_{TS}$ *versus* $p_{fusion}$ during short-term plasticity (STP) and synapse potentiation remains to be established. Evidently, not being able to separate $f_{TS}$ from $p_{fusion}$ will blur the distinction between regulation of SV priming and SV fusion (Neher, 2024).

In the present study, we combined kinetic modelling with a recently introduced analysis method suitable for differentiating between subpools of SVs in distinct priming states (Neher & Taschenberger, 2021). This allowed us to mechanistically dissect changes in synapse strength and STP in response to three different experimental manipulations that enhance transmitter release: (1) increase in $[Ca^{2+}]_i$, (2) elevation of external $[Ca^{2+}]$ or (3) stimulation of the diacylglycerol (DAG) signalling pathway. Although these manipulations lead to seemingly similar changes, that is, stronger synapses with less facilitation and more pronounced depression, we demonstrate that the underlying mechanisms are different. In particular, augmenting synaptic strength by application of a DAG mimetic and Munc13/PKC activator only moderately enhances $p_{fusion}$ but strongly increases the number of TS SVs, demonstrating that the abundance of tightly docked SVs critically determines synaptic strength and STP (Neher, 2024; Neher & Brose, 2018).

## Materials and methods

### Ethical approval

Animals were maintained at the institute's central animal facility under the supervision of an experienced specialist veterinarian for laboratory animals. Juvenile, post-hearing onset (postnatal day [P] 14–16) Wistar rats of either sex were used for experiments shown in Figs 1–8 and 9*A*. The data shown in Fig. 9*B* were obtained from Munc13-1 knock-in (D705N and D711N, referred to as Munc13[DN]) mice (Lipstein et al., 2021). Munc13[DN] mice were bred heterozygously and routinely genotyped using PCR. All experiments complied with the German Protection of Animals Act and with the guidelines for the welfare of experimental animals issued by the European Communities Council Directive. Animal health was monitored daily by caretakers and a veterinarian, and a quarterly health monitoring was done. Animals were kept at 21 ± 1°C and 55% relative humidity with a 12 h light–dark cycle. Food and tap water were provided *ad libitum*. Every effort was made to minimize the number of animals used and their suffering.

### Slice preparation

Acute brainstem slices were prepared as previously described (Lin et al., 2022). After rapid decapitation, the

brain was immediately immersed in ice-cold low-$Ca^{2+}$ and low $Na^+$ Ringer solution containing (in mM) 93 N-Methyl-D-glucamin (NMDG), 2.5 KCl, 0.5 $CaCl_2$, 25 glucose, 10 $MgCl_2$, 20 HEPES, 1.25 $NaH_2PO_4$, 30 $NaHCO_3$, 2 thiourea, 5 sodium ascorbate and 3 Na-pyruvate (pH 7.35, bubbled with 95% $O_2$, 5% $CO_2$). The brainstem was glued onto the stage of a VT1000S vibratome (Leica, Wetzlar, Germany), and ∼200 µm-thick coronal slices containing the medial nucleus of the trapezoid body (MNTB) were cut. Slices were incubated for ∼40 min at 35°C in ACSF consisting of (in mM) 125

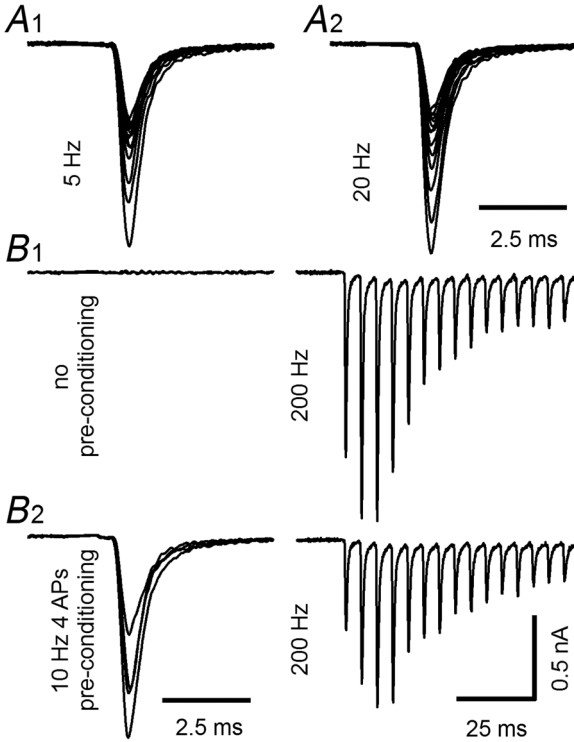

**Figure 1. Synaptic short-term plasticity (STP) in response to 5–200 Hz stimulus trains in post-hearing-onset (P14-16) rat calyx of Held synapses under conditions of physiological external $Ca^{2+}$**

*A*, Sample eEPSCs (evoked EPSCs) obtained in response to 5 Hz (A1) and 20 Hz (A2) trains consisting of 40 stimuli. For clarity, only the initial 15 eEPSCs are shown superimposed. *B*, Sample eEPSCs obtained in response to 200 Hz trains consisting of 40 stimuli (right panels). For clarity, only the initial 15 eEPSCs are shown. STP induced by high-frequency (100 and 200 Hz) trains was assayed by recording unconditioned eEPSC trains (B1) in addition to conditioned eEPSC trains (B2). The latter were preceded by two or four stimuli delivered at 10 Hz (left panel). Pre-conditioning strongly reduced the initial amplitude of the high-frequency trains but affected subsequent eEPSCs less, leading to stronger facilitation. Each trace in (*A*) and (*B*) represents an average of three repetitions. Stimulus artefacts are blanked. The bath solution contained 1.5 mM $Ca^{2+}$ and 1.5 mM $Mg^{2+}$ and was supplemented with 1 mM kyn (kynurenic acid) to minimize postsynaptic AMPAR saturation and desensitization. 5 µM strychnine was added to block IPSCs. The amplitude scale bar in (B2) applies to all panels.

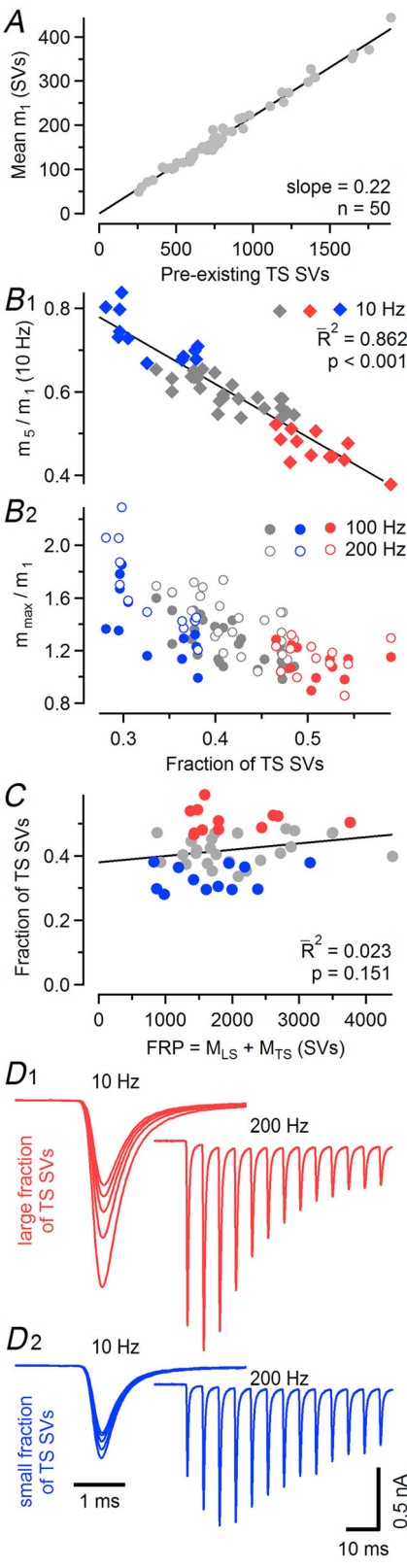

**Figure 2. Number and relative abundance of tightly docked SVs (synaptic vesicles) are key determinants of synaptic strength and short-term plasticity**

*A*, Scatter plot of mean quantal content of the initial eEPSC ($m_1$, averaged across all $f_{stim}$) *versus* number of pre-existing TS SVs ($M_{TS}$)

as obtained by NTF (non-negative tensor factorization) decomposition (Fig. S1) for all 50 synapses recorded under control conditions. Each symbol represents one synapse. The straight line represents a linear regression through the origin. The slope of the regression line is a measure of the initial $p_{fusion}$. B, Short-term depression (STD) and short-term facilitation (STF) are heterogeneous among calyx synapses and are determined by the relative abundance of LS and TS SVs. The magnitude of STD was quantified as the ratio of quantal contents of the fifth to the initial eEPSC ($m_5/m_1$) and plotted against the relative fraction of TS SVs [$f_{TS} = TS/(LS + TS)$; B1]. Each symbol represents one synapse. The 15 synapses with the strongest and the 15 synapses with the weakest depression are represented by red and blue symbols, respectively. The straight line represents a linear regression showing a strong negative correlation between the magnitude of 10 Hz STD and $f_{TS}$. The magnitude of 100- and 200 Hz STF was quantified as the ratio of the maximum eEPSC amplitude (excluding $m_1$) to the initial eEPSC amplitude ($m_{max}/m_1$) and plotted against $f_{TS}$ (B2). Each symbol represents one synapse. Matching colours identify the same synapses in B1 and B2. C, Scatter plot of $f_{TS}$ *versus* FRP (fast-releasing SV pool). The straight line represents a linear regression. D, Average eEPSC waveforms for 10 Hz (left, only the first five responses shown) and 200 Hz (right, only the first 13 responses shown) stimulation for the 15 synapses with the strongest (D1) and the 15 synapses with the weakest (D2) 10 Hz depression. ($\bar{R}^2$, adjusted coefficient of determination).

NaCl, 2.5 KCl, 2 CaCl$_2$, 1 MgCl$_2$, 25 glucose, 25 NaHCO$_3$, 1.25 NaH$_2$PO$_4$, 0.4 ascorbic acid, 3 myoinositol and 2 Na-pyruvate, which was continuously bubbled with 95% O$_2$, 5% CO$_2$ (pH 7.35). Thereafter, slices were kept at room temperature (RT, 21–24°C) and used for recordings for up to 4 h after recovery.

## Electrophysiological recordings

Whole-cell patch-clamp recordings were made from principal neurons of the MNTB at RT using an EPC-10 amplifier (HEKA Elektronik, Reutlingen, Germany, RRID:SCR_018399) controlled by Patchmaster software (HEKA Elektronik, Reutlingen, Germany, RRID:SCR_000034). Thick-walled patch pipettes made from borosilicate glass (Science Products) were coated with dental wax to minimize stray capacitance and had an open-tip resistance of 2.5–4 MΩ when filled with a Cs-gluconate based solution containing (in mM) 100 Cs-gluconate, 30 TEA-Cl, 30 CsCl, 10 HEPES, 5 EGTA, 5 Na2-phosphocreatine, 4 ATP-Mg, and 0.3 GTP, pH 7.2 with CsOH. During recordings slices were continuously perfused with normal ACSF solution containing 1.5 mM MgCl$_2$ and 1.5 mM CaCl$_2$ (control condition) supplemented with 5 μM strychnine to block glycinergic IPSCs. To reduce the amplitudes of AP-evoked EPSCs (eEPSCs) for improved voltage clamp and to attenuate postsynaptic $\alpha$-amino-3-hydroxy-5-methyl-4-isoxazolepropionic acid receptor (AMPAR) saturation and AMPAR desensitization, all experiments were performed in

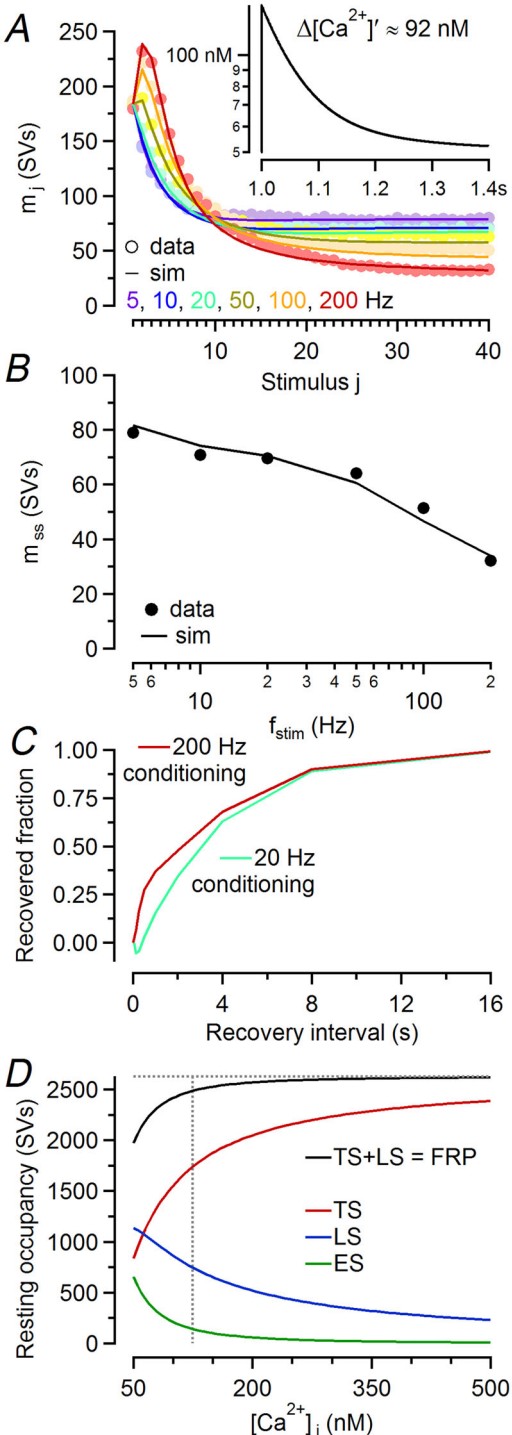

**Figure 3. A kinetic model based on a two-step priming scheme faithfully reproduces STP (short-term plasticity) in calyx synapses under conditions of physiological external Ca$^{2+}$**
A, Mean quantal contents (circles) during stimulus trains averaged over all 50 synapses assayed under control conditions and model predictions (lines) are plotted superimposed against stimulus number for regular trains consisting of 40 stimuli and $f_{stim}$ = 5–200 Hz (colour coded as indicated). The inset represents the postulated 'effective' Ca$^{2+}$ transient (semilogarithmic plot), which resembles an AP (action potential)-triggered volume-averaged global Ca$^{2+}$

transient measured in nearly unperturbed calyx terminals (Habets & Borst, 2006; Müller et al., 2007). *B*, Comparison of measured (symbols) and predicted (line) steady-state release ($m_{ss}$) for $f_{stim} =$ 5–200 Hz. $m_{ss}$ represents the mean of the last five responses for the measured EPSC trains ($m_{36}$–$m_{40}$) and $m_{40}$ for the simulations. *C*, Predicted time courses of eEPSC recovery after conditioning stimulation with 200 Hz (dark red, 40 stimuli) or 20 Hz (bright teal, 40 stimuli) trains. Note the emergence of a fast-recovery component after 200 Hz stimulation similar to experimental observations (Lipstein et al., 2013; Wang & Kaczmarek, 1998; Weingarten et al., 2022). *D*, Equilibrium occupancy of model states at rest as a function of presynaptic resting $[Ca^{2+}]_i$ between 50 and 500 nM. Half-maximum increase in the occupancy of the TS state (red) relative to resting conditions ($[Ca^{2+}]_i = 50$ nM) was achieved at $[Ca^{2+}]_i \approx 124$ nM (vertical dotted line). At resting $[Ca^{2+}]_i$ ~25% of the total number of docking sites ($N_{total}$, horizontal dotted line) were assumed to be vacant (represented by the difference between the black solid and horizontal dotted lines).

the continuous presence of 1 mM of the low-affinity GluR antagonist kynurenic acid (kyn). The blocking ratio $eEPSC_{kyn}/eEPSC_{ctrl}$ was measured in 78 calyx synapses and equal to $0.118 \pm 0.004$. For pharmacological augmentation of synaptic strength, 2.5 μM ionomycin or 1 μM phorbol 12,13-dibutyrate (PDBu) was added to the standard ACSF in some experiments. In another set of experiments, the extracellular $MgCl_2$ and $CaCl_2$ concentrations were changed from 1.5 and 1.5 mM to 1 and 2 mM, respectively.

Cells were visualized using oblique illumination (Dodt gradient contrast) through a 60× water-immersion objective (NA = 1.0, Olympus) using an upright BX51WI microscope (Olympus, Hamburg, Germany). A bipolar stimulation electrode was used to evoke presynaptic APs (stimulus intensity ≤20 V, 100 μs duration). Series resistance ($R_s$) was ≤8 MΩ and compensated ≥ 82%. Holding potential ($V_h$) and leak current were −70 mV and <300 pA, respectively. Sampling interval and low-pass filter settings were 20 μs and 5 kHz, respectively. Voltage clamp errors caused by the remaining uncompensated $R_s$ were corrected off line using a software routine similar to that described in Traynelis (1998). eEPSCs were offset corrected and digitally low-pass filtered using a cut-off frequency of 5 kHz using a Gaussian filter kernel (Colquhoun & Sigworth, 1995).

Presynaptic voltage clamp recordings from calyx of Held terminals were performed using patch pipettes with an open-tip resistance of 3.5–4.5 MΩ. Series resistance was ≤12 MΩ, and $R_s$ was compensated 60%–70%. For measuring presynaptic voltage-activated $Ca^{2+}$ currents ($I_{Ca(V)}$) and changes in membrane capacitance ($\Delta C_m$), pipettes were filled with a Cs-gluconate based solution consisting of (in mM) 100 Cs-gluconate, 30 TEA-Cl, 30 CsCl, 10 HEPES, 5 $Na_2$-phosphocreatine, 4 ATP-Mg, and 0.3 GTP, pH 7.3 with CsOH. The concentration of free calcium ($[Ca^{2+}]_i$) inside the presynaptic terminal

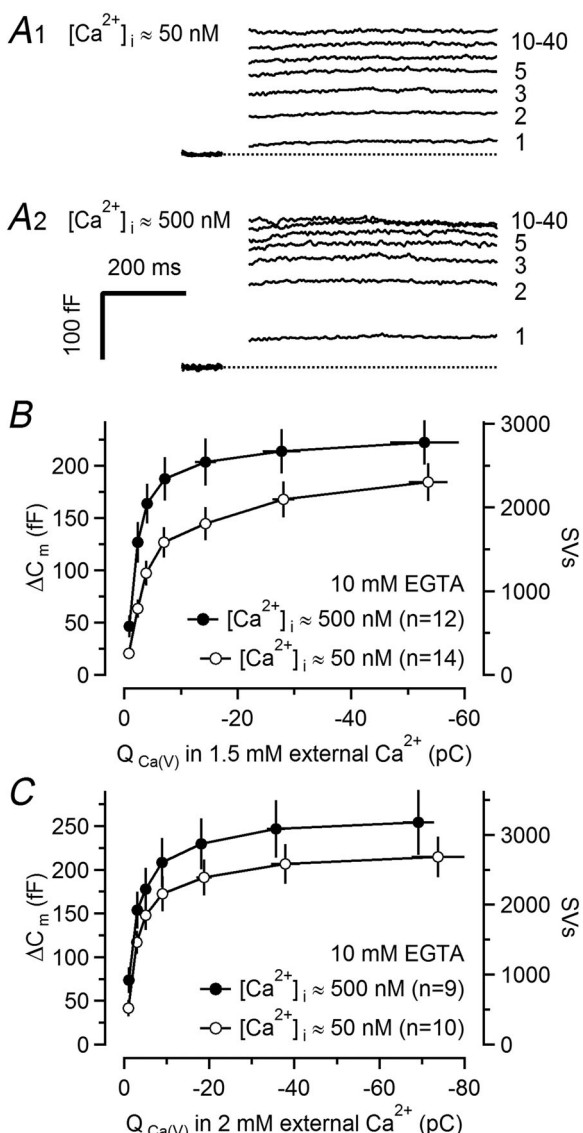

**Figure 4. Overfilling of the fast-releasing SV (synaptic vesicle) pool (FRP) at high presynaptic resting $[Ca^{2+}]_i$**
*A*, Average presynaptic $\Delta C_m$ responses elicited by depolarizing voltage steps (from $V_h = -80$ to 0 mV) of various durations (1, 2, 3, 5, 10, 20 and 40 ms, as indicated next to the traces) with either $[Ca^{2+}]_i \approx 50$ nM (A1, $n = 14$) or $[Ca^{2+}]_i \approx 500$ nM (A2, $n = 12$). $[Ca^{2+}]_i$ was adjusted by adding the appropriate total $[Ca^{2+}]$ to a pipette solution containing 10 mM EGTA (see Materials and Methods). $\Delta C_m$ responses are blanked for the first 60 ms after the onset of the step depolarizations. *B*, Summary plot of presynaptic $\Delta C_m$ (mean values measured between 500 and 600 ms after depolarization) *versus* $Q_{Ca(V)}$ in response to the step depolarization [same durations as in (*A*)] for recordings with $[Ca^{2+}]_i \approx 50$ nM (empty circles) or $[Ca^{2+}]_i \approx 500$ nM (filled circles). Note that the FRP is nearly completely depleted after a ≥10 ms-long depolarization. External $[Ca^{2+}]$ was 1.5 mM in (*A*, *B*). Note the faster saturation of $\Delta C_m$ responses with increasing step duration at $[Ca^{2+}]_i \approx 500$ nM, possibly reflecting a larger fraction of TS SVs at elevated $[Ca^{2+}]_i$. *C*, Summary plot of similar experiments as illustrated in (*A*, *B*) but in a different set of calyx terminals and using 2 mM external $[Ca^{2+}]$ (empty circles: $[Ca^{2+}]_i \approx 50$ nM, $n = 10$; filled circles: $[Ca^{2+}]_i \approx 500$

nM, $n = 9$), causing larger $Q_{Ca(V)}$. The conversion of $\Delta C_m$ to number of SVs [right axes in (*B*) and (*C*)] assumes a single vesicle capacitance of 80 aF (Sakaba, 2006). Number of terminals tested in (*B*) and (*C*) is given in parentheses.

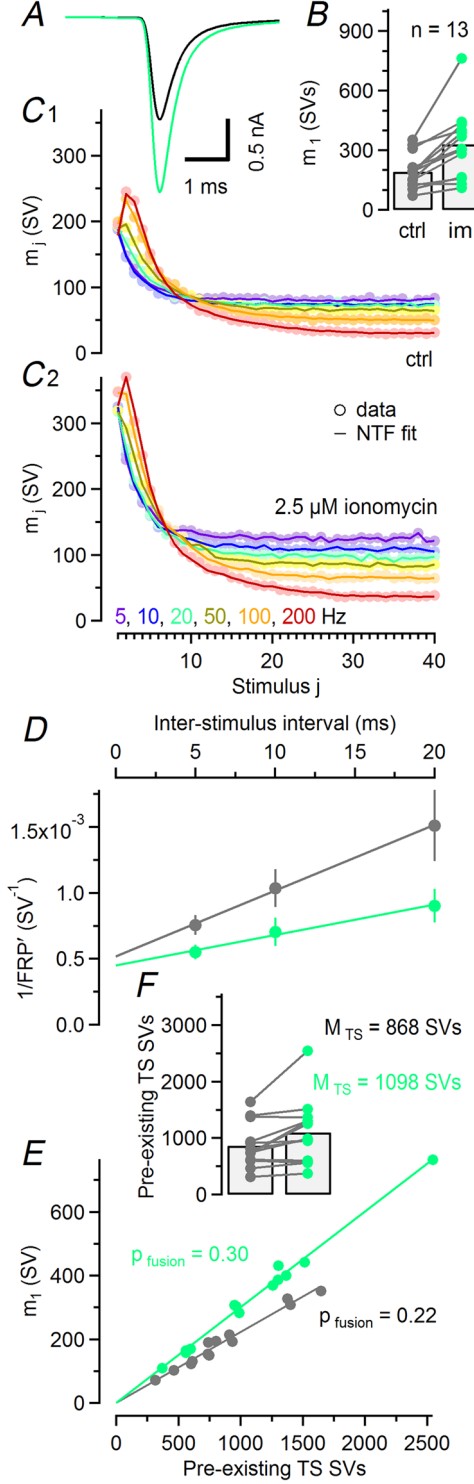

was 'clamped' using a high $Ca^{2+}$ buffer concentration. To achieve the desired $[Ca^{2+}]_i$ of $\sim$50 or $\sim$500 nM, 3.7 or 8.5 mM $CaCl_2$, respectively, was added to a pipette solution containing 10 mM EGTA. The bath solution was supplemented with 1 $\mu$M TTX, 1 mM 4-AP and 40 mM TEA-Cl to suppress voltage-activated $Na^+$ and $K^+$ currents.

All experiments were performed at RT. All off-line analysis of electrophysiological data and numerical simulations were performed using Igor Pro (WaveMetrics, Lake Oswego, OR, USA, RRID:SCR_000325).

## Decomposition of quantal release into distinct components using non-negative tensor factorization

Non-negative tensor factorization (NTF) is an iterative algorithm which can be used to decompose a given set of eEPSC train amplitudes obtained from several synapses into a sum of few components, providing estimates both for the time courses of the individual components and for their contributions to each eEPSC. NTF of eEPSC trains was performed similarly as previously described (Lin et al., 2022; Neher & Taschenberger, 2021). eEPSCs were recorded in response to afferent fibre stimulation using trains consisting of 40 stimuli delivered at frequencies ($f_{stim}$) of 5, 10, 20, 50, 100 and 200 Hz. In addition, pre-conditioned eEPSC trains were recorded for the two highest $f_{stim}$ (100 and 200 Hz) by delivering two or four stimuli at 10 Hz prior to the 40 stimuli trains. All stimulus protocols were applied under control conditions and in subsets of the synapses additionally after drug application (ionomycin or PDBu) or after switching to elevated external $[Ca^{2+}]$. A minimum of three repetitions of eEPSC trains were recorded for each condition. Slices were discarded subsequent to recordings in presence of ionomycin or PDBu because of a potentially incomplete washout of the drugs. To convert eEPSC peaks into quantal content ($m$), we assumed an 'effective quantal size' $q^* = -6.6$ pA in presence of 1 mM kyn in the bath. The quantity $q^*$ is slightly smaller than the expected mean mEPSC amplitude in presence of kyn to account for the temporal dispersion of quanta during AP-evoked synchronous release, thereby enabling us to derive quantal content estimates from eEPSC peak amplitudes without requiring deconvolution (Lin et al., 2022; Neher & Taschenberger, 2021; Taschenberger et al., 2005).

Peak amplitudes of individual EPSC train repetitions were measured for each condition, and the respective $m_j$

**Figure 5. Ionomycin-induced augmentation of synaptic strength is caused by a larger pool of TS SVs (synaptic vesicles) as well as enhanced $p_{fusion}$**
*A*, Average eEPSC waveforms obtained from 13 calyx synapses before (black) and after (mint) application of 2.5 $\mu$M ionomycin in the bath. *B*, Bar graph and dot plot represent mean and individual values, respectively, for $m_1$ before (left) and after (right) application of ionomycin. *C*, Mean quantal contents ($m_1$–$m_{40}$, circles) during

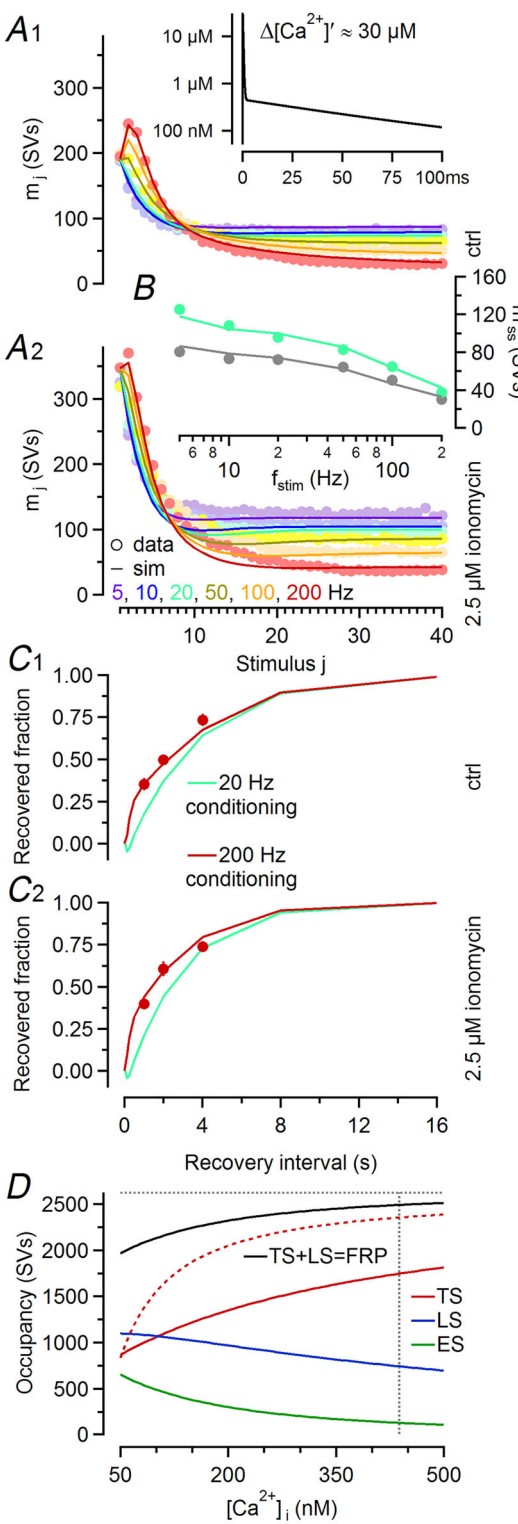

stimulus trains averaged over all 13 synapses measured before (C1) and after (C2) application of ionomycin are plotted against stimulus number ($f_{stim}$ = 5–200 Hz; colour coded as indicated). NTF (non-negative tensor factorization)-decomposition fits (lines) are shown superimposed. *D*, Scatter plots of 1/FRP' (fast-releasing SV pool) over ISI (interstimulus interval) for recordings before (grey) and after (mint) ionomycin application. Intersections of the line fits (solid lines) with the *y*-axis represent the estimates for 1/FRP corrected for incomplete pool depletion (Lin et al., 2022; Lipstein et al., 2021; Lopez-Murcia et al., 2024). *E*, Scatter plot of mean $m_1$ across all $f_{stim}$ *versus* number of TS SVs pre-existing prior to stimulation, as obtained using NTF decomposition for all 13 synapses before (grey) and after (mint) application of ionomycin. Each symbol represents one synapse. Straight lines represent linear regressions through the origin. The slopes of the regression lines are a measure of initial $p_{fusion}$. *F*, Bar graph and dot plot representing mean and individual values, respectively, for the number of pre-existing TS SVs before (left) and after (right) application of ionomycin.

values for each stimulus *j* in the train were averaged before NTF decomposition. The NTF input data derived from *N* synapses, which were stimulated at the six different $f_{stim}$, were 40 × *N* standard tensors consisting of one layer, except for $f_{stim}$ = 100 and 200 Hz, for which two additional tensor layers were obtained by pre-conditioning the synapses. During data acquisition, synapses were allowed sufficient resting time after each stimulus train to ensure complete recovery from activity-induced changes in synaptic transmission (15 s after ≤10 Hz trains, 25 s after 20 and 50 Hz trains, 40 s after 100 and 200 Hz trains). The order of the stimulus trains was pseudo-randomized for each synapse to minimize the effects of systematic run-down or run-up of synaptic responses.

During NTF decomposition, the number of quanta contributed by a given component, for example, $M_{TS}$, $M_{LS}$ and $M_{RS}$, is allowed to vary among synapses, whereas the normalized release time courses (termed 'base functions' [BFs], normalized to a cumulative sum of one) are constrained to be identical for all synapses recorded under the same experimental condition and at the same $f_{stim}$. In addition, the initial values of $BF_{TS}$ are constrained to be nearly identical across all $f_{stim}$, whereas the initial values of $BF_{LS}$ and $BF_{RS}$ remained close to zero because they were initialized with very small guess values. NTF decomposition was performed in two sequential steps. (1) First a two-component NTF decomposition was performed to derive the estimates for the number of TS SVs in individual synapses at rest ($M_{TS}$) and the two BFs, $BF_{TS}$ and $BF_{LS,RS}$, which represent the normalized time courses of release contributed by SVs that were tightly docked (TS SVs) prior to stimulation and those that were not (LS SVs and RS SVs), respectively. Both BFs were initialized as previously described (Neher & Taschenberger, 2021), and 200 iterations were performed during which the goodness of fit was tracked. (2) Subsequently a three-component NTF decomposition was

**Figure 6. Predictions of an LS–TS model with a revised Ca²⁺ sensitivity for the ionomycin-induced changes in synaptic strength and STP (short-term plasticity)**
*A*, Mean quantal contents ($m_1$–$m_{40}$, circles) during stimulus trains averaged over all 13 synapses measured before (A1) and after (A2) application of ionomycin are plotted against stimulus number ($f_{stim}$ = 5–200 Hz; colour coded as indicated). Model predictions (lines)

are shown superimposed. The inset in (A1) shows the waveform of the 'effective' $Ca^{2+}$ transient (semilogarithmic plot; compare to Fig. 3A), which represents the sum of a large and fast-decaying local and a small and slowly decaying global $Ca^{2+}$ transient contributing ~20% and ~80% to the $Ca^{2+}$ time integral, respectively. B, Steady-state release ($m_{ss}$, circles) plotted as a function of stimulation frequency for recordings obtained before (grey) and after (mint) application of ionomycin. Model predictions (lines) are shown superimposed. C, Predicted time courses of eEPSC recovery after conditioning stimulation with 200 Hz (dark-red lines, 40 stimuli) or 20 Hz (bright-teal lines, 40 stimuli) trains in the absence (C1) and presence (C2) of ionomycin. The experimentally observed fractional recovery at intervals of 1, 2 and 4 s after 200 Hz conditioning is shown superimposed (dark-red circles). D, Equilibrium occupancy of model states at rest as a function of resting $[Ca^{2+}]_i$ between 50 and 500 nM. The revised model features reduced $Ca^{2+}$ sensitivity, that is, smaller $\sigma_1$ and $\sigma_2$ values, of the priming rate constants $k_1$ and $k_2$. Half-maximum increase in the occupancy of the TS state (red) relative to resting conditions ($[Ca^{2+}]_i = 50$ nM) was achieved at $[Ca^{2+}]_i \approx 438$ nM (vertical dotted line). At a resting $[Ca^{2+}]_i$ of ~50 nM, ~25% of the total number of docking sites (horizontal dotted line) were assumed to be vacant (represented by the difference between the black solid and horizontal dotted lines). For comparison the occupancy of state TS as a function of resting $[Ca^{2+}]_i$ in Fig. 3D is shown (red dashed line).

---

carried out yielding three BFs: $BF_{TS}$, $BF_{LS}$ and $BF_{RS}$ representing the release of those SVs that were in either TS or LS state prior to stimulation and those SVs which were in neither of these two states (RS SVs), respectively. In this second step, $BF_{TS}$ was constrained to the result obtained from the initial two-component NTF decomposition, and the analysis was run for 100 iterations.

The estimate for the mean $M_{LS}$ was systematically adjusted during iterations by shifting it by 25%–35% towards a pre-determined target value in each cycle. This adjustment was necessary because the three-component NTF decomposition does not ensure a unique separation between $M_{LS}$ and $M_{RS}$ (Neher & Taschenberger, 2021). The target value was calculated as previously described (Lin et al., 2022). Briefly, our earlier work showed that due to the rapid LS→TS transition at elevated $[Ca^{2+}]_i$, the sum of TS SVs and LS SVs at rest corresponds approximately to the total number of fast-releasing SVs [FRP (fast-releasing SV pool)] in calyces (Sakaba & Neher, 2001a, b) as determined by high-frequency stimulation. The FRP size was estimated from the cumulative quantal release during high-frequency stimulation after correcting for pool replenishment and extrapolating to infinite $f_{stim}$. Because cumulative release during high-frequency trains only partially depletes the FRP (Neher, 2015), three separate apparent FRP estimates (FRP′) were obtained at $f_{stim} = 50$, 100 and 200 Hz. By plotting 1/FRP′ against interstimulus interval (ISI) and extrapolating a regression line to ISI = 0 ms, we effectively estimated the FRP for infinite $f_{stim}$ and assumed that this approach compensates for incomplete FRP depletion (Lin et al., 2022; Lipstein

et al., 2021; Lopez-Murcia et al., 2024). The mean $M_{TS}$ was then subtracted from the FRP to estimate $M_{LS}$, which represents the number of LS SVs pre-existing prior to stimulation.

### Kinetic scheme for SV priming and fusion

Time courses of AP-evoked synchronous release during stimulus trains were simulated using a kinetic scheme previously described (Lin et al., 2022; Lopez-Murcia et al., 2024). This kinetic scheme is based on the following key assumptions: (1) docking/priming of SVs occurs at a single type of release site, and the total number ($N_{total}$) of functionally identical release sites is fixed; (2) SVs docking/priming steps are reversible such that SV states maintain a dynamic equilibrium; (3) only SVs equipped with a mature release machinery, that is, in the tightly docked (TS) or labile tightly docked (TSL) state, are fusion competent; (4) the rate constants for the ES→LS transition ($k_1$) and for the LS→TS transition ($k_2$) are $Ca^{2+}$ dependent, whereas all other rate constants have fixed values; (5) immediately after an SV fusion event, release sites are in a refractory state (ERS) and become available for refilling (ES) with a docked SV with first-order kinetics ($b_4$); and (6) the refilling of vacant release sites is assumed to utilize an infinite replenishment pool.

These model properties imply that a release site at any given time $t$ can be either empty and available for docking/priming ($N_{ES}$), occupied by a docked/primed SV, or empty and unavailable for docking/priming ($N_{ERS}$). SV docking and priming proceed through two sequential maturation states: SVs first become loosely docked (LS SVs) before they mature into tightly docked fusion-competent vesicles (TS SVs). A small fraction ($\kappa$) of LS SVs transition into a labile fusion-competent state (TSL SVs) immediately after each AP. As pointed out previously (Lin et al., 2022; Neher, 2024), the TSL had to be postulated to achieve net facilitation of release at $f_{stim} = 200$ Hz, that is, EPSC paired-pulse ratios (PPRs) >1. The $Ca^{2+}$-dependent acceleration of the LS→TS transition alone is not sufficient to achieve PPRs >1, if one limits $p_{fusion}$ increases to those returned by NTF analysis. TSL SVs quickly revert to the LS state due to a high backward rate constant for the LS←TSL transition ($b_3$). Thus the entire pool of docked/primed SVs at a given time $t$ can be subdivided into LS, TS and TSL SV subpools ($SP_{LS}$, $SP_{TS}$, $SP_{TSL}$, respectively), such that

$$N_{total} = N_{ES}(t) + SP_{LS}(t) + SP_{TS}(t) + SP_{TSL}(t)$$
$$+ N_{ERS}(t) \tag{1}$$

The following coupled ordinary differential equations describe temporal changes in state occupancies at rest and during ISIs:

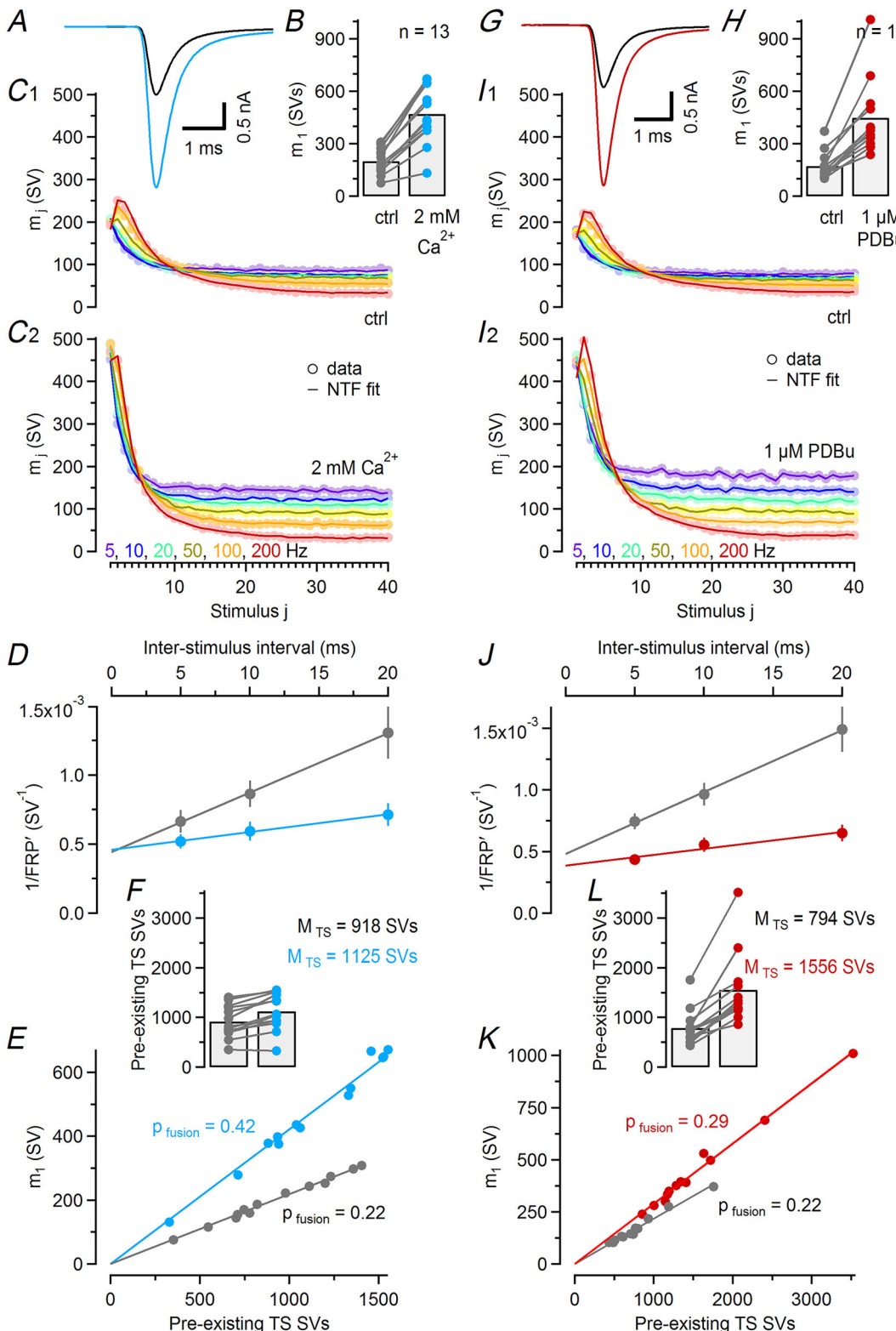

**Figure 7. Comparison of changes in synaptic strength and STP (short-term plasticity) induced by high external Ca²⁺ or PDBu (phorbol 12,13-dibutyrate) application**

*A*, Average eEPSC waveforms obtained from 13 calyx synapses before (black) and after (blue) application of 2 mM external $Ca^{2+}$ in the bath. *B*, Bar graph and dot plot representing mean and individual values, respectively, for $m_1$ before (left) and after (right) application of 2 mM external $Ca^{2+}$. *C*, Mean quantal contents ($m_1$–$m_{40}$, circles) averaged over all 13 synapses measured before (C1) and after (C2) application of 2 mM external $Ca^{2+}$

were plotted against stimulus number. NTF (non-negative tensor factorization)-decomposition fits (lines) are shown superimposed. Trains consisted of 40 stimuli ($f_{stim}$ = 5–200 Hz, colour coded as indicated). D, Scatter plots of 1/FRP' (fast-releasing SV pool) over ISI (interstimulus interval) for recordings before (grey) and after (blue) application of 2 mM external $Ca^{2+}$. Intersections of line fits (straight lines) with the y-axis represent estimates for 1/FRP corrected for incomplete pool depletion. E, Scatter plot of mean $m_1$ across all $f_{stim}$ versus number of TS SVs (synaptic vesicles) pre-existing prior to stimulation, as obtained using NTF decomposition for all 13 synapses in 1.5 mM (grey) and in 2 mM (blue) external $Ca^{2+}$. Each symbol represents one synapse. Straight lines represent linear regressions through the origin. The slopes of the regression lines (straight lines) are a measure of the initial $p_{fusion}$. F, Bar graph and dot plot representing mean and individual values, respectively, for the number of pre-existing TS SVs before (left) and after (right) application of 2 mM external $Ca^{2+}$. G–L, Similar experiments as described in (A–F) except that 1 μM PDBu was applied in the bath to augment synaptic strength while external $[Ca^{2+}]$ was kept at 1.5 mM. In presence of PDBu the average $m_1$ increased similarly as observed in 2 mM external $Ca^{2+}$ (compare B and H). Note, however, the pronounced increase in FRP size (J) and number of pre-existing TS SVs (L) but relatively small increase in $p_{fusion}$ (K).

$$\frac{d}{dt} N_{ES}(t) = -k_1 \cdot N_{ES}(t) + b_1 \cdot SP_{LS}(t) + b_4 \cdot N_{ERS} \quad (2)$$

$$\frac{d}{dt} SP_{LS}(t) = -(b_1 + k_2) \cdot SP_{LS}(t) + b_2 \cdot SP_{TS}(t)$$
$$+ b_3 \cdot SP_{TSL}(t) + k_1 \cdot N_{ES}(t) \quad (3)$$

$$\frac{d}{dt} SP_{TS}(t) = -b_2 \cdot SP_{TS}(t) + k_2 \cdot SP_{LS}(t) \quad (4)$$

$$\frac{d}{dt} SP_{TSL}(t) = -b_3 \cdot SP_{TSL}(t) \quad (5)$$

$$\frac{d}{dt} N_{ERS}(t) = -b_4 \cdot N_{ERS}(t) \quad (6)$$

Note that $SP_{TSL} = 0$ for resting synapses and after ISIs $\gg 1/b_3$. Varying the fraction of empty sites ($N_{ES}/N_{total}$) at rest in a certain range leads to similar model predictions, if $k_1$ is appropriately scaled to keep the product of $k_1 \cdot N_{ES}$ constant. Provided that this is the case, predicted release time courses are quite similar for $N_{ES}/N_{total}$ in the range of 15%–30%.

This system of coupled ordinary differential equations [eqns (2)–(6)] was solved numerically using Igor Pro's built-in routine 'IntegrateODE' and choosing a fifth-order Runge–Kutta–Fehlberg algorithm. The backward rate constants $b_1$, $b_2$, $b_3$ and $b_4$ have fixed values, and the two forward rate constants $k_1$ and $k_2$ are modelled as $Ca^{2+}$-dependent quantities which increase linearly with the 'effective' $[Ca^{2+}]_i$ according to

$$k_1(t) = k_{1,rest} + \sigma_1 \cdot \left([Ca^{2+}](t) - [Ca^{2+}]_{rest}\right) \quad (7)$$

$$k_2(t) = k_{2,rest} + \sigma_2 \cdot \left([Ca^{2+}](t) - [Ca^{2+}]_{rest}\right) \quad (8)$$

$[Ca^{2+}]_{rest}$ represents the resting $[Ca^{2+}]_i$, which was assumed to be 50 nM (Lou et al., 2005). $\sigma_1$ and $\sigma_2$ are linear slope factors characterizing the $Ca^{2+}$ dependence of the SV priming steps.

For each SV fusion event, the quantal content $m_j$ of the $EPSC_j$ triggered by stimulus $j$ was calculated as the product of $p_{fusion,j} \cdot (SP_{TS}(t_j) + SP_{TSL}(t_j))$, with subpool sizes and $p_{fusion}$ evaluated immediately before stimulus arrival. Index $j$ indicates stimulus index, $j = 1$–40. $SP_{TS}$ and $SP_{TSL}$ were decremented by their contribution to $m_j$, and $N_{ERS}$ was incremented by $m_j$. $[Ca^{2+}](t)$ was assumed to decay back to its resting value $[Ca^{2+}]_{rest}$ with a double-exponential time course:

$$[Ca^{2+}](t) = \Delta[Ca^{2+}]' \cdot (1 - f_{slow}) \cdot \exp\left(\frac{-(t - t_j)}{\tau_{fast}}\right)$$
$$+ \Delta[Ca^{2+}]' \cdot f_{slow} \cdot \exp\left(\frac{-(t - t_j)}{\tau_{slow}}\right) + [Ca^{2+}]_{rest} \quad (9)$$

where $\Delta[Ca^{2+}]'$ is the amplitude of the AP-evoked 'effective' $Ca^{2+}$ transient ($\Delta[Ca^{2+}]$) adjusted according to the dynamic changes in presynaptic $Ca^{2+}$ influx during stimulus trains as previously described (Lin et al., 2022). $f_{slow}$, $\tau_{slow}$ and $\tau_{fast}$ denote fraction and decay time constant of the slow-decaying component and decay time constant of the fast-decaying component, respectively, and $t_j$ is the onset time of the jth stimulus.

For the model with a revised $Ca^{2+}$ sensitivity of the SV priming steps, we postulated a $Ca^{2+}$ transient composed of a fast-decaying local transient ($\Delta[Ca^{2+}]l_i = 30\ \mu M$) plus a small and slowly decaying global transient ($\Delta[Ca^{2+}]g_i = 454$ nM), contributing ~16% and ~84%, respectively, to the total $Ca^{2+}$ time integral:

$$[Ca^{2+}](t) = \Delta[Ca^{2+}]'_l \cdot \exp\left(\frac{-(t - t_j)}{\tau_{fast}}\right)$$
$$+ \Delta[Ca^{2+}]'_g \cdot (1 - f_{slow}) \cdot \exp\left(\frac{-(t - t_j)}{\tau_{fast}}\right)$$
$$+ \Delta[Ca^{2+}]'_g \cdot f_{slow} \cdot \exp\left(\frac{-(t - t_j)}{\tau_{slow}}\right) + [Ca^{2+}]_{rest} \quad (10)$$

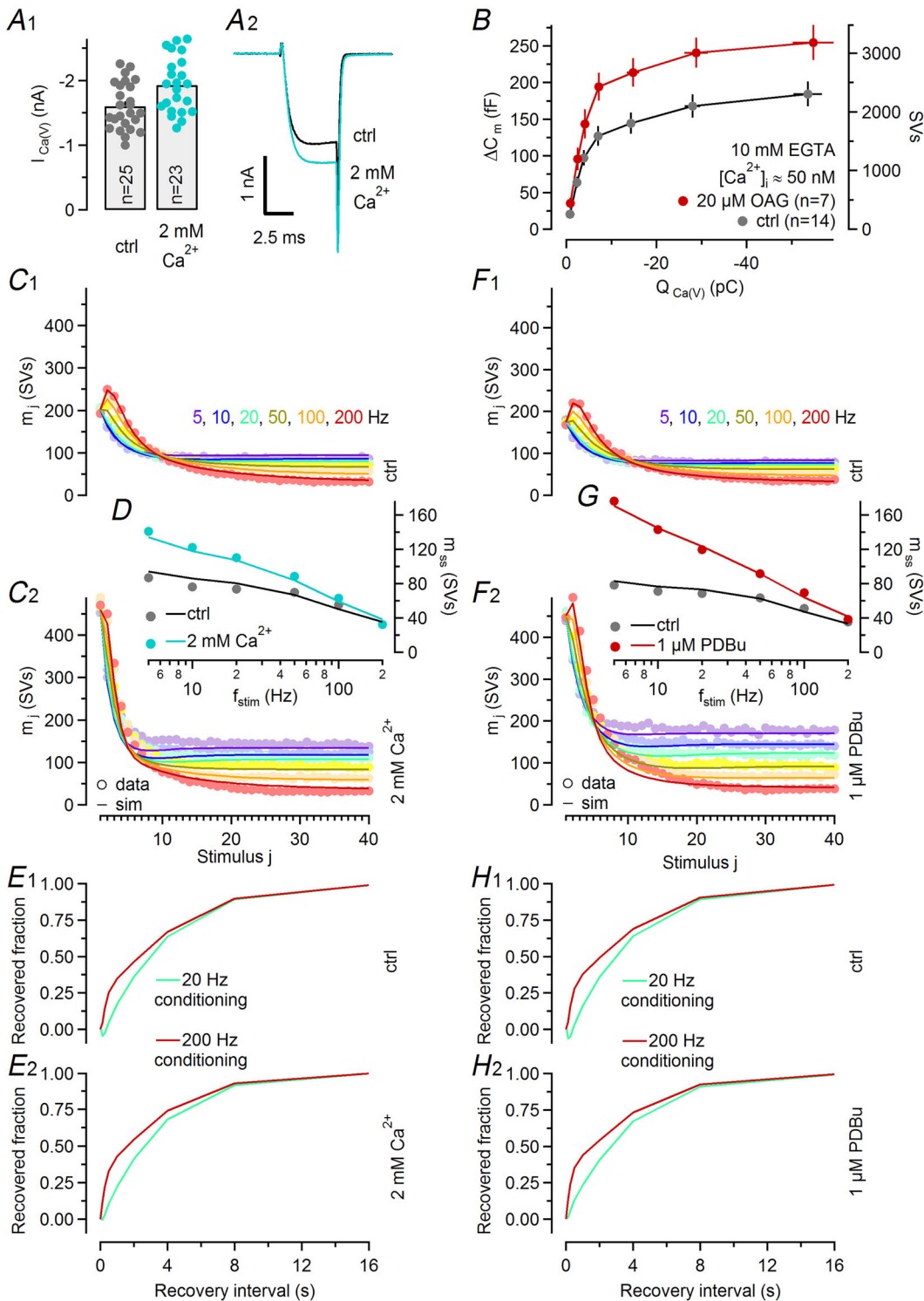

**Figure 8. Predictions of an LS–TS model with a revised Ca²⁺ sensitivity for high external Ca²⁺- and PDBu (phorbol 12,13-dibutyrate)-induced changes in synaptic strength and STP (short-term plasticity)**

*A*, Changes in presynaptic Ca²⁺ influx when increasing external [Ca²⁺]. Bar graph and dot plot represent mean and individual values, respectively, for $I_{Ca(V)}$ measured in 1.5 mM (grey) or 2 mM (blue) external [Ca²⁺] (A1). Average waveforms of presynaptic $I_{Ca(V)}$ in response to a 5 ms step depolarization from $V_h = -80$ to 0 mV are shown in the right panel (A2). *B*, Summary plot of presynaptic $\Delta C_m$ (mean value measured between 500 and 600 ms after

depolarization) *versus* $Q_{Ca(V)}$ (step depolarizations of 1, 2, 3, 5, 10, 20 and 40 ms duration) for recordings with $[Ca^{2+}]_i \approx 50$ nM in the absence (grey, same data as in Fig. 4*B*) or presence (red) of the DAG (diacylglycerol) analogue 1-oleoyl-2-acetyl-*sn*-glycerol OAG (20 µM) in the pipette solution. Number of calyx terminals tested is given in parentheses. *C*, Mean quantal contents ($m_1$–$m_{40}$, circles) during stimulus trains averaged over all 13 synapses measured first in 1.5 mM (B1) and subsequently in 2 mM (B2) external $[Ca^{2+}]$ were plotted against stimulus number ($f_{stim} = 5$–200 Hz, colour coded as indicated). Model predictions (lines) are shown superimposed. *D*, Steady-state release ($m_{ss}$) plotted as a function of stimulation frequency for recordings obtained in 1.5 mM (grey) and in 2 mM (blue) external $[Ca^{2+}]$. *E*, Predicted time courses of eEPSC recovery after conditioning stimulation with 200 Hz (dark red, 40 stimuli) or 20 Hz (bright teal, 40 stimuli) trains in 1.5 mM (E1) and in 2 mM (E2) external $[Ca^{2+}]$. *F*, Mean quantal contents (circles) averaged over all 13 synapses measured before (F1) and after (F2) application of 1 µM PDBu were plotted against stimulus number. Model predictions (lines) are shown superimposed. *G*, Steady-state release ($m_{ss}$) plotted as a function of stimulation frequency for recordings obtained before (grey) and after (red) application of 1 µM PDBu. In (*C*) and (*F*) model predictions (lines) are superimposed onto experimental data (circles). *H*, Predicted time courses of eEPSC recovery after conditioning stimulation with 200 Hz (dark red, 40 stimuli) or 20 Hz (bright teal, 40 stimuli) trains in the absence (H1) and in presence (H2) of 1 µM PDBu.

# Results

Short-term synaptic plasticity (STP) in response to regular stimulus trains ($f_{stim} = 5$–200 Hz; Fig. 1) was characterized in 50 post-hearing (P14–P16) rat calyx Held synapses recorded at nearly physiological external $[Ca^{2+}]$ (1.5 mM; Borst, 2010). Because postsynaptic AMPAR desensitization and saturation can modulate synaptic depression and/or occlude facilitation of transmitter release at glutamatergic synapses (Chanda & Xu-Friedman, 2010; Chen et al., 2002; Foster et al., 2005; Harrison & Jahr, 2003; Sun & Beierlein, 2011), including calyx of Held synapses (Habets & Borst, 2005; Taschenberger et al., 2002; Wong et al., 2003), recordings were obtained in presence of the low-affinity, rapidly dissociating AMPAR agonists kyn (1 mM). For 38 out of these 50 synapses, we additionally characterized STP after augmenting synapse strength by either (1) bath application of ionomycin (2.5 µM, $n = 13$) to elevate the presynaptic $[Ca^{2+}]_i$, (2) increasing external $[Ca^{2+}]$ (from 1.5 to 2 mM, $n = 13$) to increase AP-induced $Ca^{2+}$ influx or (3) bath application of PDBu (1 µM, $n = 12$) to increase Munc13-mediated SV priming activity (Rhee et al., 2002) (Fig. S1). Peak amplitudes of eEPSCs during stimulus trains were converted to quantal content before subjecting the data to NTF to obtain estimates for model parameters, such as SV subpool sizes and initial fusion probability ($p_{fusion}$) for each experimental condition (Lin et al., 2022; Neher & Taschenberger, 2021).

## Number and relative abundance of tightly docked SVs determine synaptic strength and STP

To correlate synaptic strength and STP characteristics under control conditions with the number and relative abundance of differentially primed SVs in individual synapses, we subjected the data from all 50 synapses tested to NTF decomposition (Fig. S1A). As adapted to the analysis of trains of synaptic responses (Neher & Taschenberger, 2021), NTF describes time courses of quantal release during stimulus trains by the sum of up to three contributions from SV populations, which had been in different functional states prior to stimulation (Lin et al., 2022; Neher & Taschenberger, 2021), that is, pre-existing LS and TS SVs, and vesicles which are newly recruited to release sites during stimulation (RS SVs). Two-component NTF splits the release time course into components contributed by SVs residing in state TS ($M_{TS}$) and those not residing in state TS ($M_{LS,RS}$) before stimulation. The initial value of $BF_{TS}$ corresponds to the initial $p_{fusion}$, and $M_{TS}$ values provide an estimate for the number of TS SVs in individual synapses at rest (Neher & Taschenberger, 2021). Three-component NTF further separates quantal release not originating from pre-existing TS SVs into components contributed by pre-existing LS SVs ($BF_{LS}$ and $M_{LS}$) and by SVs newly recruited to SV docking sites ($BF_{RS}$ and $M_{RS}$) (Neher & Taschenberger, 2021) (Fig. S2).

Three-component NTF decomposition described the mean time courses of quantal release in response to regular 5–200 Hz trains (Fig. S2A1) and to pre-conditioned 200 Hz (Fig. S2A2) and 100 Hz trains (Fig. S2A3) very well as indicated by the near-perfect correspondence between experimental data and NTF fit result. The corresponding $BF_{TS}$ and $BF_{LS}$ are shown in Figs S2B1 and S2B2. BFs for $f_{stim} = 5$–20 Hz had similar time courses, indicating that, despite different durations of ISIs, similar numbers of SVs undergo fusion (TS SVs) or make an LS→TS transition followed by vesicle fusion (LS SVs) per ISI (Neher & Taschenberger, 2021). Importantly, not only mean time courses of quantal release (Fig. S2A) but also release time courses of individual synapses are well described by the NTF decomposition fit, as illustrated by the close correspondence between fitted and measured $m_j$ values for each stimulus $j$ across all synapses analysed (Fig. S2C).

Consistent with previous studies using 2 mM external $[Ca^{2+}]$ (Lin et al., 2022; Neher & Taschenberger, 2021; Taschenberger et al., 2016), the quantal content of the initial eEPSC ($m_1$) measured at physiological external $[Ca^{2+}]$ was similar across all $f_{stim}$ for a given synapse (Fig. 1) but varied nearly 10-fold among different calyx

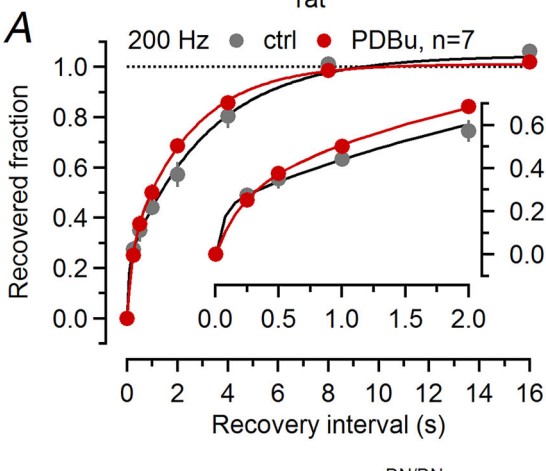

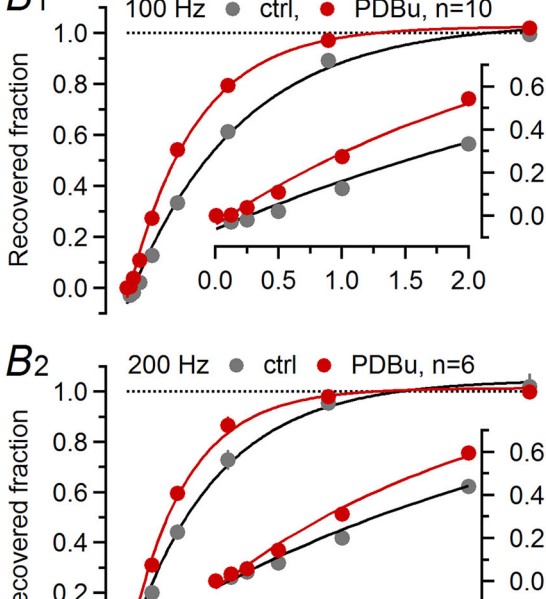

absent from Munc13$^{DN/DN}$ calyx synapses, which more clearly uncovers the acceleration of the slow recovery component. The insets in (*A*) and (*B*) illustrate the first 2 s of the eEPSC recovery time course at a faster time scale.

**Figure 9. PDBu (phorbol 12,13-dibutyrate) primarily accelerates the forward rate constants of the SV (synaptic vesicle) priming reaction**

*A*, Time course of the recovery of synaptic strength from depression induced by 200 Hz trains measured in the same rat calyx synapses under control conditions (grey) and after application of 1 μM PDBu in the bath (red). The recovered fraction was calculated as (eEPSC$_{test}$ − eEPSC$_{ss}$)/(eEPSC$_1$ − eEPSC$_{ss}$), where eEPSC$_{test}$ is the eEPSC measured after one of the seven recovery intervals tested (0.25, 0.5, 1, 2, 4, 8 and 16 s), eEPSC$_{ss}$ is the average of the last five eEPSCs of the conditioning 200 Hz train and eEPSC$_1$ is the first eEPSC of the conditioning 200 Hz train. *B*, Similar experiment as shown in (*A*) but measured in mouse calyx synapses of Munc13$^{DN}$ mice (Lipstein et al., 2021) after 100 Hz (B1) or 200 Hz (B2) conditioning (the eight recovery intervals were 0.125, 0.25, 0.5, 1, 2, 4, 8 and 16 s). The fast component of recovery during the initial 500 ms seen in (*A*) is

synapses (mean $m_1 \approx 184$ SVs, range 49–443 SVs) (Fig. 2*A*). NTF -decomposition constrains the BFs to be uniform across all synapses for a given $f_{stim}$ and experimental condition. This implies similar initial $p_{fusion}$ among all synapses, which is represented by the starting value of BF$_{TS}$. A scatter plot of $m_1$ (mean over all $f_{stim}$) *versus* NTF-derived $M_{TS}$ estimates exhibits a strong linear correlation reflecting the NTF constraint (Fig. 2*A*). A slope of a linear regression through the origin to this scatter plot, thus, is a measure of the initial $p_{fusion}$, which amounted to ∼0.22 under control conditions (external [Ca$^{2+}$] = 1.5 mM). Considering the large heterogeneity among calyx synapses with respect to $m_1$, short-term depression (STD) and short-term facilitation (STF), this may seem surprising. However Fig. 2*A*,*B* shows that such heterogeneity can be fully accounted for by differences in number and relative abundance of LS and TS SVs pre-existing prior to stimulation. The magnitude of 10 Hz STD was quantified as $m_5/m_1$, and that of high-frequency (100 and 200 Hz) STF was quantified as $m_{max}/m_1$ (with $m_{max}$ defined as the largest $m$ value among $m_2$ to $m_{40}$). Values for $m_5/m_1$ ranged from ∼0.38 to ∼0.84 across all synapses (mean $m_5/m_1 \approx 0.59$) (Fig. 2*B1*). During high-frequency stimulation 90% ($f_{stim} = 100$ Hz) and 96% ($f_{stim} = 200$ Hz) of all synapses exhibited net facilitation ($m_{max} > m_1$), and $m_{max}/m_1$ values ranged from 0.90 to 1.85 (mean $m_{max}/m_1 \approx 1.25$) and 0.86 to 2.29 (mean $m_{max}/m_1 \approx 1.41$) for 100 and 200 Hz stimulation, respectively (Fig. 2*B2*). The ratio $m_5/m_1$ strongly correlated with the relative abundance of TS SVs [$f_{TS} = $ TS/(LS + TS)], with a larger proportion of TS SVs causing stronger 10 Hz STD (Fig. 2*B1*). A similar, albeit less-pronounced, correlation was observed between STF and $f_{TS}$, with a smaller proportion of TS SVs causing stronger high-frequency STF (Fig. 2*B2*). No correlation between $f_{TS}$ and FRP was observed (Fig. 2*C*), indicating that LS- or TS-dominated synapses can be found among calyx synapses regardless of their FRP size.

In summary, heterogeneous initial synaptic strength and STP under control conditions can be fully accounted for by assuming a uniform initial $p_{fusion}$ among calyx synapses (Fig. 2*A*) yet profound synapse-to-synapse variation in the resting equilibrium of SVs in functionally distinct priming states (LS *vs.* TS; Fig. 2*B*): a large number and high proportion of pre-existing TS SVs gives rise to synapses with larger initial EPSCs, strong 10 Hz STD and very little 100 or 200 Hz STF (Fig. 2*D1*). In contrast, a small number and low relative proportion of pre-existing

TS SVs gives rise to synapses with small initial EPSCs, weak 10 Hz STD and pronounced 100 or 200 Hz STF (Fig. 2*D2*). Because the mean $f_{TS}$ was ~42% under control condition, shifting in the priming equilibrium in favour of TS can account for a mean augmentation in synaptic strength ($m_1$) of ~2.4-fold (1/0.42) without a change in $p_{fusion}$.

## A kinetic model based on a two-step priming scheme reproduces STP in calyx synapses at physiological external Ca²⁺

Next we simulated time courses of mean quantal release under control conditions (averaged over all synapses) using the two-step LS–TS priming scheme described previously (Fig. 1*Cii* in Lin et al., 2022). Similar to Lopez-Murcia et al. (2024), we initially approximated the AP-induced 'effective' Ca²⁺ transient with an instantaneous increase and a double-exponential decay using $\tau_{fast} = 60$ ms (88%) and $\tau_{slow} = 240$ ms (12%) (Fig. 3*A*, inset). Using NTF-derived estimates as initial guesses, model parameters were adjusted by trial and error, which allowed us to closely reproduce STP (Fig. 3*A*) and steady-state quantal release (Fig. 3*B*) for $f_{stim} = 5$–200 Hz.

After conditioning stimulation inducing strong synaptic depression, eEPSC amplitudes recover with an exponential time course ($\tau = 3$–5 s; von Gersdorff et al., 1997), and after high-frequency conditioning, the recovery is accelerated over several hundreds of milliseconds immediately after stimulation (Lipstein et al., 2013; Lipstein et al., 2021; Wang & Kaczmarek, 1998). Both features of the eEPSC recovery time course were reproduced by the model (Fig. 3*C*).

The Ca²⁺ dependence of the priming steps is modelled using the simple linear relationship [eqns (7) and (8)]. Whereas transient elevations in [Ca²⁺]ᵢ above [Ca²⁺]rest during and after high-frequency stimulation are responsible for enhanced SV priming causing accelerated recovery from SV pool depletion, long-lasting increases in [Ca²⁺]ᵢ above [Ca²⁺]rest change the equilibrium occupancy of the model states in resting synapses and thereby regulate synaptic strength because of increased availability of TS SVs ($f_{TS}$). Figure 3*D* shows the equilibrium occupancy of the model states for [Ca²⁺]ᵢ ranging from 50 to 500 nM. At [Ca²⁺]ᵢ = 50 nM ~25% of the docking sites are empty. Increasing [Ca²⁺]ᵢ accelerates the forward priming rate constants and therefore increases LS and TS occupancies at the expense of the ES occupancy, such that the half-maximum increase in TS occupancy [TS₅₀ ₙM + 0.5 × ($N_{total}$ − TS₅₀ ₙM), where TS₅₀ ₙM is the TS occupancy at [Ca²⁺]ᵢ = 50 nM] is achieved at [Ca²⁺]ᵢ ≈ 124 nM. At this [Ca²⁺]ᵢ the relative fraction of TS SVs $f_{TS}$ increased from ~0.42 to ~0.70, and the fraction of empty sites decreased to less than 6%.

## The pool of fast-releasing docked SVs (FRP) overfills at 10-fold elevated [Ca²⁺]ᵢ

Reversibility of SV priming, which is a key assumption of the two-step LS–TS priming scheme, implies that a fraction of all SV docking sites remain vacant even under resting conditions because the equilibrium occupancy depends on the ratio of priming ($k$) and un-priming ($b$) rate constants at rest (Neher, 2024; Neher & Brose, 2018). We assumed that ~25% of the docking sites remain empty in resting calyx synapses under control conditions. To validate this assumption experimentally, presynaptic membrane capacitance changes ($\Delta C_m$) were measured in response to step depolarizations to estimate the size of the FRP (Sakaba & Neher, 2001b), which represents the sum of LS and TS SVs (Fig. 3*D*, black trace), due to the rapid LS→TS transition at elevated [Ca²⁺]ᵢ. $\Delta C_m$ measurements at 10-fold elevated [Ca²⁺]ᵢ (~500 nM), which is expected to increase site occupancy by stimulating SV priming (Eshra et al., 2021), were compared to those obtained at normal [Ca²⁺]ᵢ (~50 nM) (Fig. 4). To 'clamp' presynaptic [Ca²⁺]ᵢ the pipette solution was supplemented with a mixture of 10 mM EGTA and the appropriate total [Ca²⁺]ᵢ (see Materials and Methods). Adding 10 mM EGTA to the pipette solution strongly limits the spatial extent of local Ca²⁺ domains in the vicinity of open voltage-gate Ca²⁺ channels (VGCCs) and prevents substantial increases in the volume-averaged global [Ca²⁺]ᵢ (Eisner et al., 2023). These recording conditions, thus, facilitate quantifying the FRP size when using long step depolarizations (Lin et al., 2011; compare to their Fig. 10*C*) by (1) preventing fusion of SVs docked at larger distances from VGCCs (slowly releasing SVs = SRP; Fedchyshyn & Wang, 2005; Sakaba & Neher, 2001b; Wadel et al., 2007) and by (2) slowing pool replenishment during and after step depolarizations (Hosoi et al., 2007; Sakaba & Neher, 2001a; Wang & Kaczmarek, 1998). $\Delta C_m$ increased quickly with increasing length of the depolarizing steps and plateaued for durations ≥20 ms (Fig. 4*B*), indicating SV pool depletion. For [Ca²⁺]ᵢ ≈ 50 nM smaller average $\Delta C_m$ values were measured at all pulse durations (Fig. 4*B*). For pulse durations of 10, 20 and 40 ms, $\Delta C_m$ was ~29%, ~21% and ~17% smaller, respectively, for [Ca²⁺]ᵢ ≈ 50 nM compared to [Ca²⁺]ᵢ ≈ 500 nM. Similar results were obtained at external [Ca²⁺] = 1.5 mM (Fig. 4*A,B*) or 2 mM (Fig. 4*C*). Assuming that 20 ms pulses cause nearly complete depletion of the FRP, we concluded that its size increased ~1.27-fold (from 2099 to 2671 SVs) when [Ca²⁺]ᵢ increased 10-fold. This FRP increase may represent an underestimate, because the rate of spontaneous SV fusion increases to ~10–20 s⁻¹ at [Ca²⁺]ᵢ ≈ 500 nM (Lou et al., 2005), which may prevent complete filling of the FRP, and is therefore in good agreement with our assumption of ~25% empty docking sites at [Ca²⁺]ᵢ ≈ 50 nM, which allows a maximum

increase in FRP of $1/0.75 = 1.33$-fold. Even though $\Delta C_m$ measurements cannot resolve LS and TS subpool sizes, the faster pool depletion kinetics at elevated $[Ca^{2+}]_i$ (the ratio of $\Delta C_m$ at 2 ms over $\Delta C_m$ at 20 ms increased from $\sim 0.38$ to $\sim 0.59$; Fig. 4$B$) is consistent with an increased $f_{TS}$ at $[Ca^{2+}]_i \approx 500$ nM.

### The SV priming process has a relatively low sensitivity to small elevations in presynaptic resting $[Ca^{2+}]_i$

At P9–12 calyx synapses small presynaptic depolarizations that elevate presynaptic resting $[Ca^{2+}]_i$ to $>100$ nM increase eEPSCs by $\sim 1.5$–$2.0$-fold (Awatramani et al., 2005). We next sought to evaluate the relative contributions of increased $p_{fusion}$ *versus* a higher abundance of TS SVs ($f_{TS}$) to this eEPSC enhancement, with the principal aim of constraining the model parameters $\sigma_1$ and $\sigma_2$ [eqns (7) and (8)] which define the dynamic equilibrium between LS and TS SV subpools depending on presynaptic $[Ca^{2+}]_i$. In a subset of 13 synapses, NTF decomposition was performed under control conditions and, additionally, after application of 2.5 µM ionomycin in the bath (Fig. S1A,B). Ionomycin application increased $m_1$ on average $\sim 1.70$-fold (from $\sim 193$ to $\sim 330$ SVs; Fig. 5$A$,$B$), consistent with an ionomycin-induced elevation in $[Ca^{2+}]_i$ to $>100$ nM (Awatramani et al., 2005). Steady-state release ($m_{ss}$) was enhanced for all $f_{stim}$ in presence of ionomycin. For example for $f_{stim} = 10$ Hz, $m_{ss}$ increased $\sim 1.48$-fold (from $\sim 73$ to $\sim 108$ SVs), whereas relative steady-state depression (SSD $= m_{ss}/m_1$) changed from $\sim 0.39$ to $\sim 0.33$ (Fig. 5$C$). In addition, STF in response to high-frequency stimulation was reduced. Mean PPR values at ISIs of 5 and 10 ms decreased from $\sim 1.26$ and $\sim 1.18$, respectively, under control conditions (Fig. 5$C1$), to $\sim 1.06$ and $\sim 0.98$, respectively, in presence of ionomycin (Fig. 5$C2$). FRP estimates obtained from the analysis of cumulative release measured during 50, 100 and 200 Hz stimulation indicated a slightly larger FRP in presence of ionomycin (Fig. 5$D$), which is consistent with the $\sim 1.12$-fold increase based on NTF decomposition (Table 1). NTF-derived estimates for $M_{TS}$ and $p_{fusion}$ increased on average $\sim 1.26$-fold (Fig. 5$F$) and $\sim 1.35$-fold (Fig. 5$E$), respectively, indicating that increased $f_{TS}$ and enhanced $p_{fusion}$ contributed approximately equally to the augmentation of synaptic strength at elevated $[Ca^{2+}]_i$.

The only $\sim 1.26$-fold increase in $M_{TS}$ in presence of ionomycin disagrees with the $\sim 1.90$-fold increase at $[Ca^{2+}]_i > 100$ nM predicted by the simulations shown in Fig. 3$D$. We therefore reduced the model parameters defining the $Ca^{2+}$ sensitivity of the priming process ($\sigma_1$ and $\sigma_2$) and postulated a larger AP-induced 'effective' $Ca^{2+}$ transient to preserve model predictions for STP despite lower $\sigma_1$ and $\sigma_2$ values. We considered

**Table 1. Summary of observed changes in mean parameter values after different experimental manipulations that increase synaptic strength**

| Parameter | Experimental conditions | | |
|---|---|---|---|
| | 1.5 mM external Ca$^{2+}$, 1.5 mM external Mg$^{2+}$, ± 2.5 µM ionomycin | 2 mM external Ca$^{2+}$, 1 mM external Mg$^{2+}$ | 1.5 mM external Ca$^{2+}$, 1.5 mM external Mg$^{2+}$, ± 1 µM PDBu |
| $m_1$ | ~1.70-fold<br>from ~193 to ~330 SVs | ~2.36-fold<br>from 200 to 472 SVs | **~2.68-fold**<br>from 172 to 449 SVs |
| FRP (NTF derived) | ~1.12-fold<br>from 1967 to 2205 SVs | ~0.97-fold<br>from 2183 to 2128 SVs | *~1.32-fold*<br>from 1942 to 2570 SVs |
| $M_{TS}$ | ~1.26-fold<br>from 868 to 1098 SVs | ~1.23-fold<br>from 918 to 1125 SVs | **~2.02-fold**<br>from 794 to 1556 SVs |
| $p_{fusion}$ | *~1.35-fold*<br>from 0.22 to 0.30 | ~1.91-fold<br>from 0.22 to 0.42 | *~1.32-fold*<br>from 0.22 to 0.29 |

eEPSC trains were first measured under control condition (1.5 mM external Ca$^{2+}$, 1.5 mM external Mg$^{2+}$) and subsequently under one of the three conditions listed here. The bath solution was always supplemented with 1 mM kynurenic acid and 5 µM strychnine. Relative changes in parameters represent the averages of the respective parameter changes in each individual synapse; changes greater than 50% or less than 50% are listed in bold or italic font, respectively. Abbreviations: FRP, fast-releasing SV pool; NTF, non-negative tensor factorization; PDBu, phorbol 12,13-dibutyrate; SV, synaptic vesicle.

**Table 2. Parameters for modelling STP during stimulus train under control conditions and after augmenting synaptic strength by ionomycin, elevated external [Ca²⁺] or PDBu**

| Model parameter | ±Ionomycin | | External [Ca²⁺] | | ±PDBu | |
|---|---|---|---|---|---|---|
| | – | 2.5 μM | 1.5 mM | 2.0 mM | – | 1 μM |
| Global $\Delta[\text{Ca}^{2+}]_i$ $\tau_{\text{fast}}$ (s) | 0.06 | 0.06 | 0.06 | 0.06 | 0.06 | 0.06 |
| Global $\Delta[\text{Ca}^{2+}]_i$ $\tau_{\text{slow}}$ (s) | 0.23 | 0.23 | 0.23 | 0.23 | 0.23 | 0.23 |
| Global $\Delta[\text{Ca}^{2+}]_i$ fraction slow | 0.15 | 0.15 | 0.15 | 0.15 | 0.15 | 0.15 |
| Global $\Delta[\text{Ca}^{2+}]_i$ amplitude (M) | 4.54e-07 | 4.54e-07 | 4.54e-07 | **5.45e-07** | 4.54e-07 | 4.54e-07 |
| Local $\Delta[\text{Ca}^{2+}]_i$ $\tau$ (s) | 0.00025 | 0.00025 | 0.00025 | 0.00025 | 0.00025 | 0.00025 |
| Local $\Delta[\text{Ca}^{2+}]_i$ amplitude (M) | 3e-05 | 3e-05 | 3e-05 | **3.6e-05** | 3e-05 | 3e-05 |
| Resting $[\text{Ca}^{2+}]_i$ (M) | 5e-08 | **1.3e-07** | 5e-08 | **9e-08** | 5e-08 | 5e-08 |
| $p_{\text{fusion}}$ | 0.22 | **0.30** | 0.22 | **0.42** | 0.22 | **0.29** |
| $N_{\text{total}}$ (SVs) | 2622 | 2622 | 2910 | 2910 | 2590 | **3297** |
| $k_{1,\text{rest}}$ (s⁻¹) | 0.370 | 0.370 | 0.374015 | 0.374015 | 0.373 | 0.373 |
| $b_1$ (s⁻¹) | 0.221 | 0.221 | 0.215 | 0.215 | 0.210 | 0.210 |
| $\sigma_1$ (M⁻¹ s⁻¹) | 2.245e+06 | 2.245e+06 | 2.181e+06 | 2.181e+06 | 2.051e+06 | 2.051e+06 |
| $k_{2,\text{rest}}$ (s⁻¹) | 0.199 | 0.199 | 0.191 | 0.191 | 0.187 | **0.378** |
| $b_2$ (s⁻¹) | 0.253 | 0.253 | 0.264 | 0.264 | 0.271 | 0.271 |
| $\sigma_2$ (M⁻¹ s⁻¹) | 1.014e+06 | 1.014e+06 | 1.042e+06 | 1.042e+06 | 1.019e+06 | **2.436e+06** |
| $\tau_{\text{TSL}} = 1/b_3$ (s) | 0.09 | 0.09 | 0.09 | 0.09 | 0.09 | 0.09 |
| $\kappa_{\text{TSL}}$ | 0.1 | **0.15** | 0.1 | 0.1 | 0.11 | 0.11 |
| $b_4$ (s⁻¹) | 2.6 | **4.2** | 2.5 | **2.9** | 2.8 | 2.8 |
| $y_{\text{inc}}$ | 0.39 | 0.39 | 0.39 | 0.39 | 0.39 | 0.39 |
| $z_{\text{dec}}$ | 0.4 | 0.4 | 0.4 | 0.4 | 0.4 | 0.4 |
| $y_{\text{max}}$ | 1.31 | **1.16** | 1.28 | 1.28 | 1.27 | 1.27 |
| $z_{\text{min}}$ | 0.87 | **1** | 0.84 | 0.84 | 0.87 | 0.87 |
| $\tau_y$ (s) | 0.017 | 0.017 | 0.017 | 0.017 | 0.017 | 0.017 |
| $\tau_z$ (s) | 3 | 3 | 3 | 3 | 3 | 3 |

For experiments in the absence and presence of ionomycin or PDBu, external [Ca²⁺] was 1.5 mM for both recording conditions. Parameter changes in comparison to the respective control conditions are indicated in bold font.
Abbreviations: PDBu, phorbol 12,13-dibutyrate; STP, short-term plasticity; SV, synaptic vesicle.

contributions of local and global Ca²⁺ signals to the activity-dependent enhancement of SV priming (Weingarten et al., 2022) and assumed an 'effective' Ca²⁺ transient composed of the sum of a local Ca²⁺ transient (amplitude = 30 μM, $\tau$ = 250 μs) and a global Ca²⁺ transient (amplitude = 454 nM, $\tau_{\text{fast}}$ = 60 ms [85%], $\tau_{\text{slow}}$ = 230 ms [15%]). The decay of the postulated global 'effective' Ca²⁺ transient is similar to that of the AP-induced volume-averaged global Ca²⁺ transient (Habets & Borst, 2006; Müller et al., 2007), but its amplitude is larger. However larger Ca²⁺ transients were indeed recorded from confocal spots on the synaptic face of calyx terminals in comparison to non-synaptic regions (Nakamura et al., 2015). The revised two-step LS–TS model with a lower Ca²⁺ sensitivity but larger 'effective' Ca²⁺ transient (Table 2) reproduces ionomycin-induced changes in synaptic strength and STP (Fig. 6A,B) and predicted eEPSC recovery after FRP depletion (Fig. 6C) reasonably well. Because the parameters $k_{1,\text{rest}}$ and $k_{2,\text{rest}}$

are unaltered, the equilibrium occupancy of the model states at [Ca²⁺]ᵢ = 50 nM is preserved. However the [Ca²⁺]ᵢ required for achieving half-maximum increase in TS occupancy relative to resting conditions more than tripled from ∼124 nM (Fig. 3D) to ∼438 nM (Fig. 6D). In agreement with $\Delta C_m$ measurements (Fig. 4B), the predicted FRP size was ∼22% smaller at [Ca²⁺]ᵢ = 50 nM compared to [Ca²⁺]ᵢ = 500 nM.

## Comparison of changes in synaptic strength and STP induced either by elevated external Ca²⁺ or by PDBu application

We proposed previously that the SVs constituting the FRP are functionally heterogeneous, and only TS SVs with a mature release apparatus are fusion competent (Lin et al., 2022; Neher & Brose, 2018). In this scenario not only $p_{\text{fusion}}$ but also the relative abundance of TS SVs ($f_{\text{TS}}$)

determine synaptic strength (Neher, 2024; Neher & Brose, 2018). To evaluate the contributions of mechanisms that target either $p_{fusion}$ or $f_{TS}$, changes in synaptic strength and STP were analysed in response to elevating external $[Ca^{2+}]$ (from 1.5 to 2 mM, $n = 13$ synapses) or after application of the DAG mimetic PDBu (1 μM, $n = 12$ synapses) (Fig. 7). For both experimental manipulations, NTF decomposition was performed on eEPSC trains recorded first under control conditions and subsequently after inducing eEPSC augmentation (Fig. S1A,B).

Elevating external $[Ca^{2+}]$ from 1.5 to 2 mM increased $m_1$ on average ∼2.36-fold (from 200 to 472 SVs; Fig. 7A,B), which is considerably stronger than the ionomycin-induced augmentation (Fig. 5A,B; Table 1). Similar to ionomycin 2 mM external $[Ca^{2+}]$ enhanced steady-state release for all $f_{stim}$ tested. For $f_{stim} = 10$ Hz for example, $m_{ss}$ increased ∼1.61-fold (from 76 to 122 SVs), whereas SSD changed from ∼0.38 to ∼0.27 (Fig. 7C). Furthermore STF in response to high-frequency stimulation was strongly reduced in presence of 2 mM external $[Ca^{2+}]$. Mean values for PPRs at ISIs of 5 and 10 ms decreased from ∼1.26 and ∼1.18, respectively, under control conditions (Fig. 7C1), to ∼1.06 and ∼0.98, respectively, in 2 mM external $[Ca^{2+}]$ (Fig. 7C2). FRP estimates based on cumulative release during 50, 100 and 200 Hz trains were nearly identical for 1.5 and 2 mM external $[Ca^{2+}]$ (Fig. 7D), consistent with the values reported by NTF decomposition (Table 1). Notably, apparent FRP size estimates (FRP′) obtained at the three individual $f_{stim}$ 50, 100 and 200 Hz were consistently larger for 2 compared to 1.5 mM external $[Ca^{2+}]$ (Fig. 7D), which was previously interpreted as a $Ca^{2+}$ dependence of the 'effective' RRP size (Thanawala & Regehr, 2013). However, pool depletion is likely far more complete at 2 compared to 1.5 mM external $[Ca^{2+}]$ at any given $f_{stim}$, such that the correction for incomplete pool depletion diminishes the apparent pool size differences (Fig. 7D). Therefore, we can conclude that the FRP remains largely unaltered when external $[Ca^{2+}]$ increased (Mahfooz et al., 2016), whereas $M_{TS}$ increased ∼1.23-fold (Fig. 7F) and $p_{fusion}$ increased ∼1.91-fold (Fig. 7E). Taken together, these data indicated that increased $p_{fusion}$ primarily accounts for enhanced transmission after elevating external $Ca^{2+}$, consistent with the expected increase in AP-triggered presynaptic $Ca^{2+}$ influx.

A different scenario emerged from the NTF decomposition of eEPSC trains from PDBu-treated synapses: whereas the increase in $m_1$ (∼2.68-fold; Fig. 7G,H) was comparable to that observed in 2 mM external $[Ca^{2+}]$, $p_{fusion}$ increased much less (∼1.32-fold; Fig. 7K) (Table 1). In contrast, $M_{TS}$ increased nearly twofold (∼2.02-fold; Fig. 7L) consistent with a PDBu-induced shift in $f_{TS}$ in resting calyx synapses. The FRP, representing the sum of LS and TS, increased

substantially after PDBu application (∼1.32-fold; Fig. 7J; Table 1). Assuming that ∼25% of all release sites ($N_{total}$) are vacant in resting synapses under control conditions, such increase in FRP would require full occupancy of all available release sites in PDBu-treated synapses ($1/∼0.75 = ∼1.33$-fold). We therefore conclude that next to a shift in the LS–TS equilibrium, PDBu moderately increased the total number of release sites.

To quantify changes in presynaptic $Ca^{2+}$ influx, we measured calyceal $I_{Ca(V)}$ and observed a ∼1.20-fold increase when elevating external $[Ca^{2+}]$ from 1.5 to 2 mM (Fig. 8A). The relationship between $Ca^{2+}$ influx ($J_{Ca}$) and external $[Ca^{2+}]$ can be approximated by a Michaelis–Menten saturation equation $J_{Ca} = J_{Ca,max} \times [Ca^{2+}]/([Ca^{2+}] + EC_{50})$ with an $EC_{50}$, the value of external $[Ca^{2+}]$ for half-maximal $Ca^{2+}$ influx, of 2.6 mM (Schneggenburger et al., 1999). This relationship predicts an increase of ∼1.19-fold, which is close to the measured ratio and consistent with a saturation of $J_{Ca}$ at increasing external $[Ca^{2+}]$ (Foster et al., 2002). Assuming that changes in the amplitudes of the 'effective' $Ca^{2+}$ transients are proportional to changes in $J_{Ca}$, we therefore increased the amplitudes of the 'effective' $Ca^{2+}$ transients in simulations by 20%.

In view of the absence of a PDBu-induced RRP increase reported for cultured mouse hippocampal neurons (Basu et al., 2007), we sought to corroborate the ∼1.32-fold increased FRP size in presence of PDBu (Fig. 7J; Table 1) and performed presynaptic $\Delta C_m$ recordings with and without the DAG analogue 1-oleoyl-2-acetyl-*sn*-glycerol (OAG, 20 μM) in the pipette solution (Fig. 8B). In presence of OAG, the $\Delta C_m$-based mean FRP estimate was ∼1.43-fold larger, which is similar to the OAG-induced ∼1.31-fold FRP increase obtained by deconvolving eEPSCs evoked by 30 ms-long presynaptic depolarizations at young calyx synapses (Fig. S4 in Lee et al., 2013). Intracellular application of OAG slightly increased calyceal $I_{Ca(V)}$ (Fig. S3) (Lou et al., 2005). A slightly enhanced presynaptic $Ca^{2+}$ influx may, in part, account for the PDBu-induced increase in $p_{fusion}$ (Fig. 7K).

Having established the principal parameter changes in synapses exposed to either elevated external $[Ca^{2+}]$ or treated with PDBu (Table 1), we then set out to reproduce changes in synapse strength and STP in numerical simulations (Fig. 8). The value of $p_{fusion}$ was set to the NTF-derived estimate. In spite of this constraint, we were able to closely reproduce the experimentally observed changes in STP (Fig. 8C) and steady-state release (Fig. 8D) when elevating external $[Ca^{2+}]$ from 1.5 to 2 mM as well as eEPSC recovery time courses (Fig. 8E) (Table 2). Similarly, PDBu-induced changes in STP (Fig. 8F), steady-state release (Fig. 8G) and eEPSC recovery (Fig. 8H) were well reproduced after adjusting $p_{fusion}$, the priming rate constants $k_{1,rest}$ and the slope parameter $\sigma_2$. In addition, a ∼1.27-fold increase in $N_{total}$ was required (Table 2).

In PDBu-treated synapses, enhanced synaptic strength is primarily a consequence of a higher abundance of TS SVs because both FRP and $f_{TS}$ increase. This increase can, in principle, be achieved either by accelerated SV priming or by decelerated SV un-priming. Although the impact of increasing forward ($k_1$ and $k_2$) or decreasing backward ($b_1$ and $b_2$) rate constants on $f_{TS}$ is similar, consequences for eEPSC recovery after SV pool depletion differ: Increasing $k_1$ and/or $k_2$ accelerates the eEPSC recovery time course, whereas decreasing $b_1$ and/or $b_2$ retards it. We therefore compared the recovery of eEPSCs from depression induced by high-frequency trains in the absence and presence of PDBu. Because postsynaptic AMPAR desensitization and saturation can expedite the recovery of eEPSCs from synaptic depression at glutamatergic synapses (Foster et al., 2002; Grabner et al., 2016), including the calyx of Held synapses (Lipstein et al., 2021; compare their Figs 6*D* and S7A), it is mandatory to minimize postsynaptic effects when studying SV pool replenishment kinetics, for example, by recording in presence of low-affinity AMPAR agonists. In agreement with observations in cultured hippocampal neurons (Chang & Mennerick, 2010; Stevens & Sullivan, 1998), we observed a PDBu-induced speed-up of the eEPSC recovery time course (Fig. 9*A*). To better resolve differences in SV replenishment speed at resting $[Ca^{2+}]_i$ in presence and absence of PDBu, we made use of a Munc13-1 knock-in mouse line with strongly impaired $Ca^{2+}$ binding to the Munc13-1 C2B domain (Fig. 9*B*). Calyx synapses of these mice lack the fast component of eEPSC recovery (Lipstein et al., 2021). A comparison of eEPSC recovery time course in the absence and presence of PDBu after 100 Hz (Fig. 9*B1*) or 200 Hz (Fig. 9*B2*) stimulation revealed a prominent acceleration of eEPSC recovery. Overall these data are consistent with the notion that PDBu primarily accelerates SV priming in the calyx of Held synapses. We cannot exclude, however, that it also stabilizes the primed SV state by decelerating the un-priming reaction (Kobbersmed et al., 2020; Weichard et al., 2023).

## Discussion

In this study, we tested whether a previously proposed sequential two-step priming model (Lin et al., 2022) is capable of reproducing (1) experimentally observed STP at the calyx of Held synapses at near-physiological external $[Ca^{2+}]$ and (2) increases in synaptic strength and alterations in STP after experimental manipulations that enhance synaptic transmission, including elevated resting $[Ca^{2+}]_i$, elevated external $[Ca^{2+}]$ and stimulation of the DAG signalling pathway. We demonstrate (1) that a refined two-step priming and fusion scheme replicates changes in synaptic strength and STP in response to experimental manipulations with a small number of plausible model parameter changes, and (2) that a

combination of NTF decomposition and state modelling allows one to separate experimentally induced changes in SV priming kinetics from those in SV fusion probability. Based on this analysis we conclude the following:

(1) The LS–TS priming equilibrium has a relatively low sensitivity to changes in resting $[Ca^{2+}]_i$. The half-maximum increase in the occupancy of release sites with TS SVs relative to resting conditions is predicted for $[Ca^{2+}]_i \approx 438$ nM.
(2) This low $Ca^{2+}$-sensitivity of the priming steps implies a large amplitude of the AP-induced 'effective' $[Ca^{2+}]_i$ transient, likely representing contributions of global and local $[Ca^{2+}]_i$ transients.
(3) The enhanced synaptic strength and stronger STD after phorbol-ester application are primarily caused by enhanced SV priming, shifting the LS–TS equilibrium and reducing the number of vacant docking sites at rest, whereas phorbol ester-induced changes in $p_{fusion}$ are smaller in comparison.
(4) Phorbol-ester application enhances the forward LS→TS transition at resting and elevated $[Ca^{2+}]_i$ causing a faster recovery of synaptic strength after pool-depleting stimuli.

### The equilibrium occupancy of functionally distinct priming states determines synaptic strength and STP

Similar to previous studies performed at 2 mM external $[Ca^{2+}]$ (Lin et al., 2022; Taschenberger et al., 2016), synaptic strength varied up to 10-fold among calyx synapses recorded at physiological $[Ca^{2+}]$, and the extent of SSD negatively correlated with initial eEPSC size. Although nearly all calyx synapses exhibited net facilitation during high-frequency stimulation at physiological $[Ca^{2+}]$, the degree of facilitation varied strongly, with weaker calyx synapses generally facilitating more strongly. Because such observations have 'traditionally' been interpreted as hinting towards a non-uniform release probability among synapses (Debanne et al., 1996; Dobrunz & Stevens, 1997; Fekete et al., 2019), it may seem puzzling that both variable STP time courses among calyx synapses and their heterogeneity with respect to $m_1$ can be well described by assuming a uniform $p_{fusion}$ for a given experimental condition. However for the two-step priming scheme, the probability for a SV of being released from the FRP by an AP ($m_1$/FRP), commonly referred to as release fraction ($F$), fusion efficiency ($f_e$) or vesicular release probability ($p_{vr}$), is given by the compound probability of an FRP SV being in state TS ($f_{TS}$) multiplied by $p_{fusion}$. For control conditions, NTF analysis reported a uniform $p_{fusion}$ of ~0.22 for all synapses, but $f_{TS}$ varied more than twofold from ~0.28 to ~0.59. Thus, the corresponding $F$ values vary from $0.28 \times 0.22 \approx 0.06$ to $0.59 \times 0.22 \approx 0.13$ among calyx synapses. To calculate

the probability for an AP triggering a SV fusion event at a given release site ($p_r = m_1/N_{total}$), the total occupancy of release sites has to be considered. We assumed 25% of the total number of release sites to be empty at rest under control conditions. Correspondingly $p_r$ values range from $0.28 \times 0.22 \times 0.75 \approx 0.05$ to $0.59 \times 0.22 \times 0.75 \approx 0.10$. Thus, a variable priming equilibrium of LS and TS SVs generates a large heterogeneity of $F$ or $p_r$ values among resting calyx synapses despite a uniform $p_{fusion}$ (Neher, 2024; Neher & Brose, 2018).

When estimating $p_{fusion}$, NTF considers EPSCs obtained from an ensemble of calyx synapses, each harbouring many release sites. This does not preclude some variability in $p_{fusion}$ among all TS SVs within a given calyx terminal, for example, due to variable coupling distances to VGCCs (Keller et al., 2015; Nakamura et al., 2015; Rebola et al., 2019). In fact, such variability may account for the gradual decline in the mean $p_{fusion}$ observed during low-frequency stimulation, because TS SVs with higher $p_{fusion}$ are more likely to be consumed earlier and remaining TS SVs will, on average, have a lower $p_{fusion}$ (Betz, 1970; Christensen & Martin, 1970).

NTF decomposition, as implemented here, postulates that only TS SVs but not LS SVs are fusion competent. Although this presumption simplifies analysis, it is conceivable that some LS SVs may fuse right away in response to presynaptic APs, that is, without requiring an LS→TS transition, albeit with a lower $p_{fusion}$ compared to TS SVs. In fact, NTF decomposition works well even if the initial value of $BF_{LS}$ is not constrained to zero (Neher & Taschenberger, 2021). Nevertheless, here we postulate that LS SVs are the functional correlate of loosely docked SVs as identified using interferometric scattering microscopy and *in vitro* membrane fusion assays (Witkowska et al., 2020, 2021) and ultrastructural data (Chang et al., 2018; Fernandez-Busnadiego et al., 2013; Jung et al., 2021), and that LS SVs strictly require an LS→TS transition to become fusion competent.

Exposing calyx synapses to elevated external [Ca$^{2+}$] or PDBu more than doubled synaptic strength and strongly reduced or even abolished STF. Despite seemingly similar changes in mean $m_1$ and STP, the underlying mechanisms of synaptic augmentation differ: elevated external [Ca$^{2+}$] primarily increased $p_{fusion}$ by ~91%, whereas $M_{TS}$ increased only by ~23% and FRP size remained nearly unchanged. Because $f_{TS}$ also increased from 0.42 to 0.53, $F$ increased from $0.22 \times 0.42 \approx 0.09$ to $0.42 \times 0.53 \approx 0.22$ at 2 mM external [Ca$^{2+}$], which is consistent with 'traditional' $F$ estimates for calyx of Held synapses at comparable developmental stage and experimental conditions ($F = 0.13–0.2$; Koike-Tani et al., 2008; Taschenberger et al., 2002, 2005).

Phorbol-ester treatment primarily increased $M_{TS}$ by ~102%, consistent with the larger fraction of tightly docked SVs (<5 nm from the AZ membrane) in PDBu-treated cerebrocortical synaptosomes as found using cryo-electron tomography (Papantoniou et al., 2023). In contrast, $p_{fusion}$ increased only by ~32%. Because $f_{TS}$ increased from ~0.41 to ~0.61, $F$ increased from $0.22 \times 0.41 \approx 0.09$ to $0.29 \times 0.61 \approx 0.18$. Even though the latter $F$ value is slightly smaller than that obtained at 2 mM external [Ca$^{2+}$], comparable changes in $m_1$ were observed at elevated external [Ca$^{2+}$] and after PDBu application because of the ~32% larger FRP in presence of PDBu.

## Multiple presynaptic actions of DAG mimetics

At many synapses, DAG analogues such as phorbol 12,13-diacetate (PDAc), PDBu or phorbol myristate acetate (PMA) enhance synaptic transmission by targeting DAG-binding C1 domains of presynaptic proteins (Hori et al., 1999; Malenka et al., 1986; Malinow et al., 1988; Oleskevich & Walmsley, 2000; Parfitt & Madison, 1993; Waters & Smith, 2000; Yawo, 1999). In cultured hippocampal neurons expressing a DAG binding-deficient Munc13 variant, eEPSC augmentation by PDBu is nearly completely lost, which identifies Munc13 as a PDBu target in these neurons (Betz et al., 1998; Rhee et al., 2002) in addition to PKC (Wierda et al., 2007). Whereas mEPSC amplitude distributions remain largely unaltered after phorbol-ester treatment (Finch & Jackson, 1990; Wu & Wu, 2001), RRP size may increase (Chang et al., 2010; Stevens & Sullivan, 1998) and/or presynaptic APs may trigger the fusion of a larger fraction of the RRP due to an increased apparent affinity of the release machinery to Ca$^{2+}$ (Basu et al., 2007; Lou et al., 2005; Wu & Wu, 2001). Further an awakening of 'dormant' presynaptic terminals was found to contribute to PDBu-induced eEPSC enhancement in cultured hippocampal neurons (Chang et al., 2010).

Consistent with previous observations in neocortical layer 5 pyramidal cell (PC) synapses and hippocampal CA1 PCs to O-LM cell synapses (Aldahabi et al., 2024; Weichard et al., 2023), NTF decomposition of eEPSC trains recorded in PDBu-treated calyx synapses as well as numerical simulations indicates that PDBu application primarily increased the abundance of TS SVs. The PDBu-induced FRP increase of ~50% at calyx synapses falls within the range of previous reports for cultured rat hippocampal neurons (~65% or ~63% when assayed using sucrose application or FM1-43 uptake, respectively, Stevens & Sullivan, 1998; ~28% when assayed by sucrose application, Chang et al., 2010). An ~50% FRP increase is larger than what would be expected if all existing release sites become occupied. Given that only ~25% of the docking sites remain vacant in resting calyx synapses under control conditions, a shift to 100% occupancy would yield a maximum FRP increase of only $1/{\sim}0.75 \approx 1.33$-fold. Therefore, we had to postulate that PDBu increases $N_{total}$ by ~27%, presumably by recruiting

additional Munc13 protein to the AZ membrane. This notion rests on the assumption that the availability of Munc13 is a limiting factor for the build-up of functional SV docking sites (Augustin et al., 1999; Sakamoto et al., 2018; Siksou et al., 2009). Interestingly, model fits to PDBu-induced changes in synaptic strength and STP at hippocampal PC to O-LM synapses suggested a similar increase in $N_{total}$ (29%; Aldahabi et al., 2024).

PDBu treatment, furthermore, increased $p_{fusion}$ by ~32%. In part this may stem from a small increase in presynaptic $Ca^{2+}$ influx because phorbol-esters increase somatic $I_{Ca(V)}$ in a variety of neurons (Agopyan et al., 1993; Swartz, 1993; Yang & Tsien, 1993). Among recombinantly expressed VGCCs both $\alpha_{1B}$ (N-type) and $\alpha_{1E}$ (R-type) channels exhibit a 30%–40% increase in peak currents after exposure to phorbol esters, whereas $\alpha_{1A}$ (P/Q-type) channels are unaffected (Stea et al., 1995). Because P/Q-type VGCCs mediate the vast majority of AP-evoked release at post-hearing calyx synapses (Iwasaki & Takahashi, 1998), a PDBu-induced increase in calyceal $I_{Ca(V)}$ is expected to be small.

When recordings in presence of PDBu were compared with control recordings, we measured an approximately twofold increased steady-state release in response to 10 Hz stimulation and an acceleration of the recovery of eEPSCs after pool-depleting stimuli. These observations are indicative of increased priming rate constants at resting $[Ca^{2+}]_i$ ($k_{1,rest}$ and $k_{2,rest}$), which agrees well with an approximately twofold larger steady-state current in cultured hippocampal neurons in response to hyper-osmotic stimuli, which empty the *RRP* through a $Ca^{2+}$-independent release mechanism (Rosenmund & Stevens, 1996). To reproduce changes in synaptic strength and steady-state release in simulations, $k_{2,rest}$ had to be increased ~2.02-fold, whereas $k_{1,rest}$, $b_1$ and $b_2$ remained unchanged. The value of $\sigma_2$, which determines the slope of the linear relationship between $k_2$ and the 'effective' $[Ca^{2+}]_i$, had to be increased ~2.39-fold. Thus activation of the DAG pathway stimulates priming at resting $[Ca^{2+}]_i$, thereby increasing the relative abundance of TS SVs. In addition the DAG pathway strongly accelerates the LS→TS transition at elevated $[Ca^{2+}]_i$, after presynaptic activity.

### The identity of the presynaptic 'effective' $Ca^{2+}$ signal which mediates $Ca^{2+}$-dependent SV priming

Numerous molecular perturbations, particularly those affecting (1) Munc13s, (2) synaptotagmins (Syts) and (3) AZ scaffold proteins, which may interact with Syts and Munc13s (Gundelfinger et al., 2015), interfere with the activity-dependent acceleration of SV priming, while leaving basal synaptic transmission often unaffected. Syts and Munc13s bind phospholipids in a $Ca^{2+}$-dependent manner, which makes them well suited to regulate,

possibly in concert, activity-dependent SV docking and priming (Silva et al., 2021).

Genetic interference with the $Ca^{2+}$-dependent stimulation of Munc13-1 by either a loss of $Ca^{2+}$-calmodulin (CaM) binding to its CaM-binding domain (Lipstein et al., 2013; Sakaba & Neher, 2001a) or impaired $Ca^{2+}$ binding to its C2B domain (Lipstein et al., 2021) slows SV priming at calyx synapses, whereas enhanced phospholipid binding to the Munc13 C2B domain accelerates eEPSC recovery from depression (Lipstein et al., 2021). Members of the Syt protein family sense $Ca^{2+}$ via their C2A and C2B domains. Syt3 and Syt7 have a high apparent $Ca^{2+}$ affinity compared to other Syt isoforms (Sugita et al., 2002). Calyx synapses and cerebellar climbing fibre to Purkinje cell synapses of Syt3 KO mice exhibit slower recovery of eEPSCs from depression (Weingarten et al., 2022). Genetic loss of Syt7 slows the time course of recovery from SV pool depleting stimuli in cultured hippocampal neurons (Liu et al., 2014), accelerates recovery at cerebellar PC→DCN synapses in Syt7 knockout mice (Turecek et al., 2017) and does not affect recovery at calyx synapses (Weingarten et al., 2022). In cerebellar mossy fibre to granule cell synapses of Bassoon KO mice, STD during sustained high-frequency trains is enhanced, and fast recovery of eEPSCs from synaptic depression is reduced (Hallermann et al., 2010). Genetic loss of Piccolo, which binds $Ca^{2+}$ via its C2A domain, impairs the fast component of eEPSC recovery from synaptic depression in endbulb synapses (Butola et al., 2017). Genetic deletion of intersectin 1 or acute interference with intersectin function inhibits fast SV replenishment at calyx synapses (Sakaba et al., 2013; Yang et al., 2025) as do actin depolymerization by latrunculinA treatment (Sakaba & Neher, 2003) or interference with actin dynamics signalling pathways (Keine et al., 2022).

Overall, these experimental observations suggest a co-existence of several pathways by which activity-dependent increases in presynaptic $[Ca^{2+}]_i$ regulate SV priming. Because of differences in $Ca^{2+}$ binding properties and subcellular localization of the respective $Ca^{2+}$ sensor proteins, it is conceivable that local and global $Ca^{2+}$ transients jointly contribute to the regulation of SV priming. How exactly the spatio-temporal profile of presynaptic $Ca^{2+}$ transients couples to the SV priming activity furthermore depends on the details of $Ca^{2+}$ influx, buffering and extrusion (Eisner et al., 2023) as well as on AZ topography. Because of the number of uncertainties involved, including a possible saturation of the SV priming reaction, we opted here for the simplified scheme of an 'effective' $Ca^{2+}$ transient to simulate $Ca^{2+}$-dependent SV priming numerically. Based on the ionomycin-induced augmentation of eEPSC and changes in STP, we conclude that the LS–TS priming equilibrium has a relatively low sensitivity to changes in resting $[Ca^{2+}]_i$, which suggests that the amplitude of the

AP-induced 'effective' $[Ca^{2+}]_i$ transient must be large, likely representing contributions of both global and local $[Ca^{2+}]_i$ transients (Weingarten et al., 2022). The shape of the 'effective' $[Ca^{2+}]_i$ transient may be interpreted as the time course of the $Ca^{2+}$-bound fraction of $Ca^{2+}$ sensors, possibly weighted by their relative contribution to activity-dependent SV priming. Assuming that $Ca^{2+}$ binding and unbinding are fast, the $Ca^{2+}$ sensors rapidly equilibrate with $[Ca^{2+}]_i$ and increase, and decay time constants of the 'effective' $[Ca^{2+}]_i$ transient are expected to be similar to those of the AP-induced presynaptic $Ca^{2+}$ transient (Dittman & Regehr, 1998; Weis et al., 1999). If either $Ca^{2+}$ equilibration or the subsequent interaction of the $Ca^{2+}$ sensor with the release apparatus occurs more slowly, the shape of the 'effective' $[Ca^{2+}]_i$ transient may resemble a low-pass filtered version of the AP-induced presynaptic $Ca^{2+}$ transient (Dittman et al., 2000). NTF decomposition yielded similar shapes for $BF_{TS}$ and $BF_{LS}$ for all $f_{stim}$ in the range of 5–20 Hz, which is consistent with similar fractions of TS SVs fusing per AP and similar fractions of LS SVs making a LS→TS transition per ISI at such $f_{stim}$. These results indicate that binding and unbinding of presynaptic $Ca^{2+}$ to its sensor may be complete within single ISIs $\geq 50$ ms.

### Assigning functional roles to presynaptic proteins requires correct separation of SV priming and fusion

It is not uncommon that several, sometimes opposing, functions are attributed to presynaptic proteins with respect to their role in regulating transmitter release and STP. For example for Munc13s, synaptotagmins and complexins, partly conflicting results regarding their regulation of the resting *RRP* size, SV replenishment kinetics and SV fusion probability or kinetics have been obtained. Although it is conceivable that genetic and pharmacological manipulations targeting presynaptic proteins interfere with more than one step of the SV cycle, it is equally possible that an assignment of conflicting functions results from an insufficient distinction between priming and fusion reactions due to limitations in the analysis of synaptic responses (Lin et al., 2022; Neher, 2024). We studied the effect of three different manipulations, increasing $[Ca^{2+}]_i$, elevating external $[Ca^{2+}]$ or stimulating the DAG pathway, on synapse strength and STP. Not limiting the analysis to changes in initial eEPSC sizes, PPRs and SSD but combining NTF decomposition of entire eEPSC trains with kinetic modelling reveals more detailed and mechanistically different insights. NTF decomposition exploits more of the information on synapse function than what is available in mean eEPSC trains alone. Specifically, an NTF decomposition fit not only describes mean time courses of quantal release but also accounts for the variability among

all synapses in a set of recordings obtained at a given experimental condition. We expect that comprehensive modelling of larger data sets in response to diverse stimulus pattern, such as used in this study, will allow a more precise separation of changes in SV priming kinetics from those mediated by changes in $p_{fusion}$ for which NTF can provide robust estimates.

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

## Additional information

### Data availability statement

The data that support the findings of this study are archived in a public repository (https://doi.org/10.5281/zenodo.14995827). Any other data are available upon reasonable request from the authors.

### Competing interests

None declared.

### Author contributions

E.N., K.-H.L. and H.T. developed the concept. N.L. generated and validated Munc13-1$^{DN}$ mice. K.-H. L., N.L. and H.T. performed electrophysiological recordings. K.-H.L., N.L., E.N. and H.T. analysed and interpreted the data. E.N. and H.T. supplied software routines for data analysis. H.T. and M.R. performed numerical simulations. N.B. and E.N. provided conceptual input and advice. H.T. and M.R. wrote the manuscript. All authors edited and agreed to the final version of the manuscript.

### Funding

This work was supported by the German Research Foundation (Cluster of Excellence EXC 2067 'Multiscale Bioimaging' to E.N.; Excellence Strategy – EXC-2049–390688087 to N.L.; CRC 1286 'Quantitative Synaptology' to E.N. and N.L).

### Acknowledgements

The authors thank Ina Herfort for excellent technical assistance, Astrid Kanbach and Nadine Asmus for mouse husbandry, the AGCT laboratory for genotyping and Dr Francisco José López-Murcia for insightful comments on the manuscript.

### Keywords

calyx of Held, numerical simulation, short-term plasticity, synaptic transmission, synaptic vesicle fusion, synaptic vesicle priming

### Supporting information

Additional supporting information can be found online in the Supporting Information section at the end of the HTML view of the article. Supporting information files available:

**Peer Review History**
**Figure S1**
**Figure S2**
**Figure S3**

