## [Peer Review History · The Journal of Physiology]

Number and relative abundance of synaptic vesicles in functionally distinct priming states determine synaptic strength and short-term plasticity

Kun-Han Lin, Mrinalini Ranjan, Noa Lipstein, Nils Brose, Erwin Neher, and Holger Taschenberger

DOI: 10.1113/JP286282

Corresponding author(s): Holger Taschenberger (taschenberger@mpinat.mpg.de)

Review Timeline:

Submission Date:	22-Nov-2024
Editorial Decision:	18-Dec-2024
Revision Received:	02-Feb-2025
Accepted:	20-Feb-2025

Senior Editor: Katalin Toth

Reviewing Editor: Samuel Young

Transaction Report:

Dear Dr Taschenberger,

Re: JP-RP-2024-286282 "Number and relative abundance of synaptic vesicles in functionally distinct priming states determine synaptic strength and short-term plasticity" by Kun-Han Lin, Mrinalini Ranjan, Noa Lipstein, Nils Brose, Erwin Neher, and Holger Taschenberger

Thank you for submitting your manuscript to The Journal of Physiology. It has been assessed by a Reviewing Editor and by 2 expert referees and we are pleased to tell you that it is acceptable for publication following satisfactory revision.

REVISION CHECKLIST:

We look forward to receiving your revised submission.

Yours sincerely,

Katalin Toth
Senior Editor
The Journal of Physiology

REQUIRED ITEMS

- Author photo and profile. First or joint first authors are asked to provide a short biography (no more than 100 words for one author or 150 words in total for joint first authors) and a portrait photograph. These should be uploaded and clearly labelled together in a Word document with the revised version of the manuscript. See Information for Authors for further details.

- You must start the Methods section with a paragraph headed Ethical approval (https://jp.msubmit.net/cgi-bin/main.plex?form_type=display_requirements#methods).

Research must comply with The Journal's policies regarding animal experiments (<https://physoc.onlinelibrary.wiley.com/hub/animal-experiments>) and adherence to these policies must be stated in the manuscript.

Authors should confirm in their Methods section that their experiments were carried out according to the guidelines laid down by their institution's animal welfare committee, including an ethics approval reference number. The Methods section must contain a statement about access to food, water and housing, details of the anaesthetic regime: anaesthetic used, dose and route of administration, and method of killing the experimental animals.

- Your manuscript must include a complete Additional Information section, including competing interests; funding; author contributions and acknowledgements.

- Please upload separate high-quality figure files via the submission form.

- Please ensure that any tables are editable and in Word format, and wherever possible, embedded in the article file itself.

- Please ensure that the Article File you upload is a Word file.

- Your paper contains Supporting Information of a type that we no longer publish, including supplementary tables and figures. Any information essential to an understanding of the paper must be included as part of the main manuscript and figures. The only Supporting Information that we publish are video and audio, 3D structures, program codes and large data files. Your revised paper will be returned to you if it does not adhere to our Supporting Information Guidelines.

- A Data Availability Statement is required for all papers reporting original data. This must be in the Additional Information

section of the manuscript itself. It must have the paragraph heading 'Data Availability Statement'. All data supporting the results in the paper must be either: in the paper itself; uploaded as Supporting Information for Online Publication; or archived in an appropriate public repository. The statement needs to describe the availability or the absence of shared data. Authors must include in their statement: a link to the repository they have used, or a statement that it is available as Supporting Information; reference the data in the appropriate section(s) of their manuscript; and cite the data they have shared in the References section. Whenever possible, the scripts and other artefacts used to generate the analyses presented in the paper should also be publicly archived. If sharing data compromises ethical standards or legal requirements then authors are not expected to share it, but must note this in their statement. For more information, see our Statistics Policy.

- Please include an Abstract Figure file, as well as the Figure Legend text within the main article file. The Abstract Figure is a piece of artwork designed to give readers an immediate understanding of the research and should summarise the main conclusions. If possible, the image should be easily 'readable' from left to right or top to bottom. It should show the physiological relevance of the manuscript so readers can assess the importance and content of its findings. Abstract Figures should not merely recapitulate other figures in the manuscript. Please try to keep the diagram as simple as possible and without superfluous information that may distract from the main conclusion(s). Abstract Figures must be provided by authors no later than the revised manuscript stage and should be uploaded as a separate file during online submission labelled as File Type 'Abstract Figure'. Please also ensure that you include the figure legend in the main article file. All Abstract Figures should be created using BioRender. Authors should use The Journal's premium BioRender account to export high-resolution images. Details on how to use and access the premium account are included as part of this email.

EDITOR COMMENTS

Reviewing Editor:

This manuscript uses modeling in combination with experimental manipulations to analyze how phorbol esters or increased calcium to determine how they regulate specific SV lifecycle steps to impact synaptic strength. Both reviewers found the data and findings to be highly impactful, rigorous and support the conclusions drawn. Both reviewers pointed out that conclusions drawn by the model could be strengthened. This can be done by using the model to analyze previous published data generated by this group and to rule out there is an equally valid way of fitting the data with heterogenous pFusion. Finally, the authors need to carefully revise and rewrite their manuscript given the positive and careful comments by the reviewers.

Please also see 'Required Items' above.

REFeree COMMENTS

Referee #1:

Authors refined the STP model that they have previously proposed, and then demonstrated the usefulness of NTF-based differentiation of changes in vesicular fusion prob and TS vesicle fraction by applying their methods to characterizing high external Ca²⁺ and PDBu-induced enhancement of EPSC1 .

Authors noticed that previous model parameters are not compatible with the effect of ionomycin on STP with regard to Mts. They resolved the incompatibility in half-maximal [Ca²⁺] required for Mts increase between two conditions by assuming that the LS-to-TS transition senses effective local [Ca²⁺] as well as global [Ca²⁺].

Furthermore, authors characterized the synaptic mechanisms underlying enhancement of EPSC1 by high external [Ca²⁺] and PDBu, showing that the acceleration in LS-to-TS transition rate underlies PDBu-induced synaptic enhancement, while an increase in P_{fusion} does for high Ca²⁺-induced EPSC1 enhancement.

Experiments were well designed and the results support their main conclusions. To make authors' conclusions more convincing, however, following points would better to be addressed.

1. pages 10, 16 and Fig. 4:

Previously (Lin et al., *pnas*, 2023) and in this Ms. authors argued that the 'FRP' size calculated from cumulative plot of EPSCs corresponds to TS plus LS vesicles. Conventionally the 'FRP' term has been used for describing results from the deconvolution analysis under the dual patch experiments in the presence of 0.5 mM EGTA (Sakaba and Neher, 2001). Authors should make it clear whether the FRP measured from cumulative EPSCs is equal to the FRP estimates from deconvolution analysis. If they are not equal, readily releasable pool (RRP) instead of FRP may be more appropriate term for the estimates from cumulative release. Please note that not only FRP but also SRP are recruited to vesicle release during high frequency train at least in immature calyx synapses probed by comparing the effects of latrunculin and CaMip (Fig. 5 in Lee et al, *pnas*, 2012). Moreover, similar difference in the latrunculin and CaM inhibitor effects was observed in the mature calyx synapses too.

2. Fig. 4:

1) It should be noted that AP-induced EPSC1 is sensitive to 10 mM EGTA in the complex type synapses (Fig. 5 in Fekete , 2019). It is possible for the proportions of simple vs complex type synapses between two groups under comparison to affect the conclusion of un-paired experiments such as Fig. 4.

2) Please discuss whether your new model supports recruitment of all the LS vesicles in 40 ms depolarization in the presence of 10 mM EGTA. How much local and global $[Ca^{2+}]$ increase do you expect? Under this Ca^{2+} increase k_2+b_2 is sufficiently high for conversion of LS vesicles to TS ones?

3. Fig 3C, Fig 6C and Fig. 8E:

These figures show that the model predicts the acceleration of EPSC recovery kinetics after high frequency stimulation, consistent with previous reports. If possible, it would make the proposed model more convincing if authors overlap previous experimental data on these figures by reproducing from previous works as authors cited (Wang & Kaczmarek, 1998; Lipstein et al., 2013; Lipstein et al., 2021) or performing new experiments (not obligatory though)

4. Fig. 7D looks as if there is little heterogeneity in the FRP size between individual synapses. I wonder if this result suggests that the FRP size is not a key factor determining Fts (= Mts/FRP). It would be helpful to show a plot of FRP size vs. Mts. It should be noted that the RRP size was proportional to the complexity of calyx structure (Fig. 1 of Fekete et al. 2019)

Minor points:

1. $[Ca^{2+}]$ required for half maximal increase in the TS occupancy is not consistent in this Ms. 438 nM in the text but 617 nM in the legend of figure 6 and Discussion.

2. date -> data in page 19

3. Page 18, corroborated -> corroborate

Referee #2:

In this study, Lin and colleagues explore experimental manipulations that boost synaptic transmission using the framework of a quantitative presynaptic vesicle pool model at the calyx of Held. Specifically, the authors examine the impact of increased calcium influx as well as phorbol esters on synaptic transmission and short-term plasticity using a multi-state kinetic model that distinguishes between two pre-fusion states (LS and TS for loosely docked and tightly docked states, respectively). They conclude that shifts in the balance of TS and LS could account for phorbol-ester induced enhancement of synaptic transmission in addition to the broad diversity of basal synaptic strengths in a population of synapses. In contrast, enhanced calcium influx predominantly increased the probability of fusion with a minor impact on the TS/LS balance. Additionally, phorbol ester treatment increased the SV pool size as estimated using fits to their model. This work is the latest in a series of studies that have reassessed fundamental presynaptic parameters by combining this two-docked-state kinetic model with a computational data fitting method (non-negative tensor factorization). And in agreement with previous papers from this group, the general theme has been to reassign effects traditionally ascribed to vesicle fusion probability to an upstream docking step. Specifically in this study, much of the phorbol ester-induced enhancement of synaptic strength could be accounted for by a shift from the LS to the TS state rather than an increase in SV fusion probability. This interpretation is novel, interesting, and has implications for mechanistic hypotheses underlying phorbol ester enhancement. I have several questions for the authors mainly for clarification purposes.

1. One of the early and important points made in this manuscript comes from the observation that the broad variety of synaptic strengths can arise from a population of synapses with essentially identical fusion probabilities but distinct distributions of SVs among the pre-fusion states. If one were to model the synaptic responses to stimulus trains while keeping the TS/LS ratio fixed but allowing for variation in fusion probability, could your kinetic model and NTF approach generate a 'best fit' to the data that is still reasonably good or does it do a far worse job of fitting the data? This may have been done in a previous publication - I was hoping for some objective demonstration that the data is much better described by a fixed pFusion and variable TS/LS distribution than a fixed TS/LS distribution and a variable pFusion parameter. Perhaps clarifying this point at the outset will help bolster confidence that there is not an equally valid way of fitting the data with heterogeneous pFusion accounting for the range of synaptic strengths.

2. One synaptic parameter that has always been difficult to compare across preps and methods is the vesicle pool size. The authors use NTF approaches with their kinetic state model to estimate the fast-releasing SV pool and conclude that phorbol esters increase this pool by around 27%. Numerous past studies have made strong claims about the lack of an effect of phorbol esters on vesicle pool size (e.g. Lou et al 2005 for a calyceal example). Can the authors clarify how they are thinking about these disparities? Is this a matter of methodological differences? Do the authors think that there is a distinct mechanism involved in increasing the FRP compared to the shift from LS to TS? Or are both of these effects likely to derive from the same underlying biochemical changes to Munc13 for instance? And in cases where synapses are assessed using a hypertonic sucrose challenge (other preps and investigators), do the authors believe that both the LS and TS states contribute to the total pool or is this more of a measure of the TS state? Naïve readers may need some help in trying to compare the FRP described here to various distinct methods such as long depolarizing presynaptic steps or sucrose.

3. The claim that 25% of the release sites are empty in the basal state seems important but nontrivial to demonstrate. And Figure 4 could use a bit more labeling to help readers navigate the data. How critical is this value of 25% to the authors modeling and conclusions? If that number was 10% or 0%, would the resulting fits be far worse or is this a relatively minor perturbation? I think the manuscript could use some clarification to better describe both the significance of the ES magnitude and the experimental evidence supporting the proposed value.

4. What exactly is the TSL state and why is it necessary here? The logic for this particular bit of complexity was not at all clear and left me wondering if you could equally well make all the same points without the TSL state. It appears that a 50% change to the value of a parameter associated with the TSL state was required to fit the ionomycin data, but no alterations of parameters associated with the TSL state were necessary when external Ca or PDBu treatment were modeled.

5. Minor comment on Figures 3 and 4 - provide more labeling. For instance, in Fig 3A, perhaps you could eliminate the DeltaCa trace (which isn't adding much) and provide a color legend for stim frequency if that is what the different colors stand for. This shows up in several later figures but may be sufficient just to show a legend once. Some labeling in Figure 4A could help as well.

END OF COMMENTS

February 2, 2025

Dr. Katalin Tóth
Senior Editor
The Journal of Physiology

Manuscript Revision

“Number and relative abundance of synaptic vesicles in functionally distinct priming states determine synaptic strength and short-term plasticity” - Lin et al.

Dear Dr. Tóth,

We would like to sincerely thank the editors and particularly the reviewers for taking the time to assess our manuscript JP-RP-2024-286282.

We are grateful to the editors and the reviewers for allowing us to submit a revised version of our manuscript. We took their detailed criticism very seriously and performed additional experiments and simulations. We sincerely hope that our revisions to the manuscript and our arguments outlined below will address all issues raised by the reviewers, so that they can recommend our manuscript for publication.

Point-by-Point Response (black) to the Editor's and Reviewers' Comments (*blue italic*)

Reviewing Editor:

This manuscript uses modeling in combination with experimental manipulations to analyze how phorbol esters or increased calcium to determine how they regulate specific SV lifecycle steps to impact synaptic strength. Both reviewers found the data and findings to be highly impactful, rigorous and support the conclusions drawn. Both reviewers pointed out that conclusions drawn by the model could be strengthened. This can be done by using the model to analyze previous published data generated by this group and to rule out there is an equally valid way of fitting the data with heterogenous pFusion. Finally, the authors need to carefully revise and rewrite their manuscript given the positive and careful comments by the reviewers.

Please also see 'Required Items' above.

We thank the editor for the positive assessment of our work. In summary, the revised version of the manuscript contains the following changes:

1. The following required items that were previously missing are now included in this submission:

- Author photo and profile
- Ethical approval section
- Additional Information section
- Data Availability Statement
- Abstract Figure file plus Abstract Figure Legend

2. Following the reviewers' suggestion, we added a new panel C to Fig. 2. This panel demonstrates the absence of a correlation between the fraction of TS SVs (f_{TS}) and *FRP* size. In other words, TS-dominated (strongly depressing) and LS-dominated (less strongly depressing and often transiently facilitating) synapses can be found among calyx synapses of any *FRP* size.

3. Following the reviewers' suggestion, we added the experimentally observed fractional recovery at intervals of 1, 2 and 4 s in Fig. 6C (control data and in the presence of ionomycin). The measured recovery is very close to model predictions.
4. Following the reviewers' suggestion, we provide more labeling on Figs. 3, 4, 5, 6, 7 and 8.
5. We revised the wording in several places of the manuscript to facilitate readability.
6. Supplemental Figures 1–3 contain information which is not absolutely essential for the understanding of the paper. These figures were still uploaded as 'supplemental figure files' together with the revised version of the manuscript. Furthermore, they will be archived in a public repository (zenodo.org). The final link to the archived files on zenodo.org will be provided upon formal acceptance of the manuscript.
7. Supplemental Tables 1–4 are essential for the understanding of the paper. They were therefore added to the main manuscript after combining Tables 2–4 into a single table.

Referee #1:

Authors refined the STP model that they have previously proposed, and then demonstrated the usefulness of NTF-based differentiation of changes in vesicular fusion prob and TS vesicle fraction by applying their methods to characterizing high external Ca^{2+} and PDBu-induced enhancement of EPSC1 .

Authors noticed that previous model parameters are not compatible with the effect of ionomycin on STP with regard to Mts. They resolved the incompatibility in half-maximal $[Ca^{2+}]$ required for Mts increase between two conditions by assuming that the LS-to-TS transition senses effective local $[Ca^{2+}]$ as well as global $[Ca^{2+}]$.

Furthermore, authors characterized the synaptic mechanisms underlying enhancement of EPSC1 by high external $[Ca^{2+}]$ and PDBu, showing that the acceleration in LS-to-TS transition rate underlies PDBu-induced synaptic enhancement, while an increase in P_{fusion} does for high Ca^{2+} -induced EPSC1 enhancement.

Experiments were well designed and the results support their main conclusions. To make authors' conclusions more convincing, however, following points would better to be addressed.

We were very glad to read Referee #1's very positive assessment of our work.

1. pages 10, 16 and Fig. 4:

Previously (Lin et al., pnas, 2023) and in this Ms. authors argued that the 'FRP' size calculated from cumulative plot of EPSCs corresponds to TS plus LS vesicles. Conventionally the 'FRP' term has been used for describing results from the deconvolution analysis under the dual patch experiments in the presence of 0.5 mM EGTA (Sakaba and Neher, 2001). Authors should make it clear whether the FRP measured from cumulative EPSCs is equal to the FRP estimates from deconvolution analysis. If they are not equal, readily releasable pool (RRP) instead of FRP may be more appropriate term for the estimates from cumulative release. Please note that not only FRP but also SRP are recruited to vesicle release during high frequency train at least in immature calyx synapses probed by comparing the effects of latrunculin and CaMip (Fig. 5 in Lee et al, pnas, 2012). Moreover, similar difference in the latrunculin and CaM inhibitor effects was observed in the mature calyx synapses too.

The size of the *FRP* obtained from cumulative EPSCs measurements, corrected for pool replenishment and for incomplete pool depletion, was ~ 2000 SVs (this manuscript) or ~ 2100 (Lin et al., 2022) for P14–16 calyx synapses. This is similar to *FRP* estimates obtained by deconvolution which amounted to $0.5 \cdot 3550 = 1775$ SVs (Sakaba and Neher, 2001) or 1500–1700 SVs (Sakaba, 2006) or 1400 SVs (Lee et al., 2012), when taking into account that the latter three *FRP* estimates pertain to younger (P8–10) calyx synapses and that the *FRP* size increases during development (Yang et al., 2021). In fact, for rat calyx synapses $\geq P14$, the *FRP* obtained by deconvolution was >2500 SVs (Yang et al., 2021).

We completely agree with this reviewer that *SRP* SVs may contribute to release during strong stimuli, especially if total release is estimated from EPSC integrals or by deconvolution analysis. However, our *FRP* estimates are derived from EPSC peaks to which *SRP* SVs contribute only very little (Sakaba, 2006).

Importantly, as detailed already previously in the Suppl. Fig. 5 of Lin et al. (2022), the *FRP* size (LS SVs + TS SVs) was correctly predicted when simulated EPSC trains were subjected to the same *FRP* analysis procedure as used for experimental data, even when very different LS/TS ratios were assumed.

In order to avoid ambiguity and confusion, and to remain consistent with previous publications, we therefore kept the term *FRP*.

2. Fig. 4:

1) It should be noted that AP-induced EPSC1 is sensitive to 10 mM EGTA in the complex type synapses (Fig. 5 in Fekete, 2019). It is possible for the proportions of simple vs complex type synapses between two groups under comparison to affect the conclusion of un-paired experiments such as Fig. 4.

We completely agree with this reviewer, that the SV pool depletion kinetics during sustained presynaptic depolarizations is expected to be somewhat slower in the presence of 10 mM EGTA in the pipette solution as compared to lower buffer concentrations. To test this experimentally, we compared the amount of release with either 100 μ M EGTA or 10 mM EGTA in the presynaptic pipette solution in a new set of recordings. In agreement with earlier studies (Fedchyshyn and Wang, 2005; Keller et al., 2015) and with the expectations of this reviewer, we found that release evoked by AP-like step depolarizations is sensitive to EGTA in P14–16 rat calyx synapses. Specifically, 10 mM EGTA reduced the mean ΔC_m response induced by AP-like depolarizations (1 ms step) by $\sim 30\%$ (**Fig. R1**) which is close to P16–P18 mouse calyces (Fedchyshyn and Wang, 2005; their Fig. 2D).

Figure R1, Reduced exocytosis induced by 1 ms presynaptic step depolarizations in the the presence of 10 mM EGTA in the pipette solution.

A, Calyceal $I_{Ca(V)}$ induced by 1 ms step depolarizations from $V_h = -80$ mV to 0 mV. Average current waveform obtained from 6 (100 μ M EGTA, red) and 10 (10 mM EGTA) individual terminals.

B, Corresponding mean ΔC_m responses

This is, however, of little concern for the experiments illustrated in Figs. 4 and 8B. To estimate the *FRP* size, we applied longer step depolarizations which deplete the entire *FRP* (see also **Fig. R3** and related discussion). This is supported by the observation that 20 ms and 40 ms steps consistently yielded nearly identical ΔC_m values in all presynaptic recordings with 10 mM EGTA in the pipette.

2) Please discuss whether your new model supports recruitment of all the LS vesicles in 40 ms depolarization in the presence of 10 mM EGTA. How much local and global $[Ca^{2+}]$ increase do you expect? Under this Ca^{2+} increase $k_2 + b_2$ is sufficiently high for conversion of LS vesicles to TS ones?

In order to rapidly deplete both LS and TS SVs during 40 ms step depolarizations in model calculations, the following three conditions have to be met: (i) the SV fusion rate constant has to be high, (ii) the LS \rightarrow TS transition rate has to be high, and (iii) the ES \rightarrow LS transition rate has to be low. In the following section, we would like to discuss why these conditions are met in our ΔC_m recordings.

Before going into details, we would like to point out that our model describes SV priming dynamics and SV fusion during AP-evoked release and for unperturbed terminals. One cannot easily expect that release induced by sustained presynaptic depolarizations during presynaptic whole-cell recordings with 10 mM EGTA in the pipette is equally well predicted. In particular, the whole-cell recording configuration may disrupt presynaptic signaling pathways regulating the kinetics of SV priming and un-priming. Moreover, the addition of 10 mM EGTA strongly alters the spatial profile of local Ca^{2+} domains in the vicinity of open VGCCs (**Fig. R2**) and the accumulation of global $[Ca^{2+}]_i$ in the presynaptic cytosol, which in turn may alter SV release and Ca^{2+} -dependent SV recruitment.

Figure R2, Spatial profile of the local steady-state $[Ca^{2+}]_i$ in the vicinity of open Ca^{2+} channels is strongly altered by the addition of 10 mM EGTA to the presynaptic pipette solution.

A, Steady state $[Ca^{2+}]_i$ plotted as a function of distance from a Ca^{2+} channel if either no buffer is present (gray) or in the presence of one of the mobile buffer species parvalbumin (PV; blue), EGTA (red) or BAPTA (green). The presynaptic Ca^{2+} buffering conditions in unperturbed calyx terminals were approximated by assuming 200 μM PV binding sites (Müller et al., 2007). Calculations were performed according to the linear buffer approximation (LBA) (Naraghi and Neher, 1997; Neher, 1998; Bauer, 2001), using eqn. 9 in Neher (1998).

B, Similar plot as shown in **A** but using logarithmic scaling for the y-axis.

C, The ratio of $[Ca^{2+}]_i$ in the presence of 200 μM PV binding sites over that in the presence of 10 mM EGTA plotted as a function of distance from a Ca^{2+} channel. At a distance of ~200 nm (corresponding to about four SV diameters), nearly 60fold higher $[Ca^{2+}]_i$ is predicted in the presence of PV as compared to the EGTA-supplemented pipette solution.

For the calculations in **Fig. R2**, the association rate constants (k_{on}) were $3.1e6 M^{-1} \cdot s^{-1}$, $1.05e7 M^{-1} \cdot s^{-1}$ and $4.4e8 M^{-1} \cdot s^{-1}$ for PV, EGTA and BAPTA, respectively. Dissociation constants (K_D) were $1.6e-7 M$, $7.1e-8 M$ and $2.2e-7 M$ for PV, EGTA and BAPTA, respectively (Nägerl et al., 2000; Eisner et al., 2023). The diffusion coefficient for Ca^{2+} and for all three buffer species was $2.2e-10 m^2 \cdot s^{-1}$ (Naraghi and Neher, 1997). Concentrations of Ca^{2+} binding sites were 200 μM, 10 mM and 10 mM for PV, EGTA and BAPTA, respectively. The single channel current was set to 0.3 pA. For these parameters, the length constants λ (the mean distance that Ca^{2+} ions diffuse before they are captured by a buffer molecule) were 520 nm, 35 nm and 6.4 nm for PV, EGTA and BAPTA, respectively. In the presence of 200 μM PV binding sites (**Fig. R2 blue**), $[Ca^{2+}]_i$ is therefore similar to the un-buffered case (gray) for distances $r \ll 520$ nm. In contrast, λ is more than tenfold shorter in the presence of 10 mM EGTA. Thus, $[Ca^{2+}]_i$ approaches the equilibrated case already for distances $r \gg 35$ nm in the presence of 10 mM EGTA (**Fig. R2 red**). Binding of Ca^{2+} to endogenous immobile buffers is not

expected to change the spatial profile of the local steady-state $[Ca^{2+}]_i$. For simplicity, binding of Ca^{2+} to free PV and to free ATP was not considered here. Both buffers are predominantly in the Mg^{2+} -bound form.

The second priming step (LS→TS) is assumed to represent an ‘in-place transition’ of already docked SVs rendering these SVs fusion competent. Because of the tight coupling between VGCCs and SV docking sites in mature calyx synapses (Fedchyshyn and Wang, 2005; Wang et al., 2008; Kochubey et al., 2009), this step should be only minimally affected by 10 mM EGTA.

When estimating the impact of 10 mM EGTA on the first priming step (ES→LS), one needs to consider that SVs can be recruited to empty docking sites from an extended cytoplasmic volume surrounding AZ Ca^{2+} channels. **Fig. R2** shows that $[Ca^{2+}]_i$ has decayed to $\sim 1.55 \mu M$ already at $r = 79$ nm (less than two SV diameters) in the presence of 10 mM EGTA (whole-cell recording). In contrast, in the presence of 200 μM PV binding sites (unperturbed terminal), $[Ca^{2+}]_i$ has decayed to $1.55 \mu M$ only at $r = 370$ nm. Accordingly, the volume of the hemisphere which surrounds a Ca^{2+} channel and for which $[Ca^{2+}]_i \geq 1.55 \mu M$ holds is larger by a factor of $(370 / 79)^3 \approx 100$ in the presence of PV as compared to 10 mM EGTA. Therefore, many SVs that could potentially be recruited to empty docking sites are likely uncoupled from the Ca^{2+} influx in the presence of 10 mM EGTA. In conclusion, the ES→LS priming step is expected to occur at much slower speed in the presence of 10 mM EGTA as compared to its speed in unperturbed calyx terminals.

With these considerations in mind, we conducted numerical simulations of SV fusion induced by 40 ms long step depolarizations (**Fig. R3**).

Figure R3, Numerical simulation of SV fusion during a sustained presynaptic depolarization as applied during presynaptic ΔC_m recordings.

The time courses in **A-E** assume a depolarization of the presynaptic calyx terminal from -80 mV to 0 mV for a duration of 40 ms, starting at $t = 0.4$ s.

A, Simulated time courses of fusion rate constant γ (**A1**, red), the effective $[Ca^{2+}]_i$ (**A1**, blue), and the forward rate constants k_1 , k_2 , k_κ (**A2**). All traces shown in **A1,2** are low-pass filtered to account for the finite gating kinetics of VGCCs. Brief Ca^{2+} tail currents occurring during the repolarization following the 40 ms step depolarization are neglected here.

B, Simulated time courses of SV fusion rate (*solid*) and cumulative release (*dashed*) in response to a 40 ms long presynaptic step depolarization. Total release amounted to 1732 SV which corresponds to 85% of the number of SVs in the *FRP* at rest (1171 LS SVs + 859 TS SVs).

C, Comparison of the experimentally measured time course of changes in presynaptic membrane capacitance (ΔC_m , *gray*) and simulated time course of cumulative vesicle release (*dashed black*). Note that rapid vesicle endocytosis during the 40 ms depolarization may possibly contribute to the measured ΔC_m time course. Measured and predicted release are shown using independent y-axes to facilitate comparison of the depletion time courses and because we cannot expect perfectly matching *FRP* sizes in the simulations (based on fiber-stimulation-derived model parameters) and the presynaptic ΔC_m recordings.

D, Time courses of state occupancies during the simulations shown in **B**. The subpool of TS SVs is rapidly depleted from a resting occupancy of 859 SVs to 2 SVs (>99% depletion). The subpool of LS SVs is depleted from a resting occupancy of 1171 SVs to 279 SVs (~76% depletion).

E, Same simulation as shown in **D** but state occupancies are monitored for a longer time period. The subpool of LS SVs has nearly completely recovered ~3-4 s after the 40 ms depolarization while the subpool of TS SVs requires ~16 s for complete recovery. The simulation assumes that presynaptic $[Ca^{2+}]_i$ rapidly returns to resting values after the 40 ms depolarization in the presence of 10 mM EGTA as previously demonstrated (Lin et al., 2012; their Fig. 4).

For a numerical simulation of release during sustained presynaptic depolarizations (**Fig. R3**), the fractions p_{fusion} and κ had to be converted into the corresponding rate constants γ (SV fusion rate constant) and k_κ (rate constant for the LS→TSL transition).

The fusion rate constant γ was approximated as follows: Peak release rates during AP-evoked release are ~750 SVs/ms. With a subpool of ~860 TS SVs, this corresponds to a rate constant γ of ~870 s⁻¹ (**Fig. R3 A1**, red). We assumed here that γ during a sustained depolarization to 0 mV is similar to the peak release rate constant during a brief AP-evoked release transient.

The rate constant k_κ for the LS→TSL transition was approximated as follows: Our simulations indicate that during 200 Hz stimulation, the subpool of LS SVs is depleted to a low occupancy of ~240 SV at steady-state (when measured immediately before AP arrival). With a fraction κ of ~0.1 and p_{fusion} of ~0.22, the contribution of TSL SVs to steady state release is ~5.2 SVs of a total m_{ss} of ~32.6 SVs, while TS SVs contribute 32.6 SVs - 5.2 SVs ≈ 27.4 SVs. Thus, the contribution of TSL SVs to m_{ss} is ~20% of the contribution of TS SVs to m_{ss} . Because TS and TSL SVs are both derived from the LS subpool and because p_{fusion} is identical for TS and TSL SVs, we conclude that $k_\kappa \approx 0.2 \cdot k_2$ (**Fig. R3 A2**, purple).

With an effective $[Ca^{2+}]_i$ of 30 μ M during the depolarization, the LS→TS transition rate constant k_2 was calculated as $k_2 = k_{2,rest} + \sigma_2 \cdot 30 \mu\text{M} \approx 30 \text{ s}^{-1}$ (**Fig. R3 A2**, blue). As detailed above, the ES→LS transition rate constant k_1 was assumed to be much less accelerated during the step depolarization $[Ca^{2+}]_i$: $k_1 = k_{1,rest} + \sigma_1 \cdot 0.01 \cdot 30 \mu\text{M} \approx 1 \text{ s}^{-1}$ (**Fig. R3 A2**, red).

Fig. R3 C illustrates that numerical simulations based on the LS-TS model with properly adjusted model parameters reproduce experimental ΔC_m responses reasonably well. During presynaptic step depolarizations in the presence of 10 mM EGTA, a rapid SV fusion rate (γ) together with a high k_2 values which is about 30times higher than that of k_1 lead to >99% depletion of the TS subpool and ~76% depletion of the LS subpool.

3. Fig 3C, Fig 6C and Fig. 8E:

These figures show that the model predicts the acceleration of EPSC recovery kinetics after high frequency stimulation, consistent with previous reports. If possible, it would make the proposed model more convincing if authors overlap previous experimental data on these figures by reproducing from previous works as authors cited (Wang & Kaczmarek, 1998; Lipstein et al., 2013; Lipstein et al., 2021) or performing new experiments (not obligatory though)

When simulating experimental data, we tried to ascertain that not only initial synaptic strength and STP during trains of 6 different stimulus frequencies is correctly reproduced but also the EPSC recovery time course is close to experimental observations. It is not uncommon for studies in this field to be limited to modeling 1 or 2 stimulation frequencies and/or paired-pulse ratios only. According to our experience, inspecting the model predictions for EPSC recovery guards against choosing non-plausible parameter value combinations. This helped us already in previous studies to constrain model parameters (Lopez-Murcia et al., 2024; their Fig. S4).

The studies referred to by this reviewer measured recovery in mouse synapses (we used rats) and at different external $[Ca^{2+}]$.

Routinely measuring the EPSC time courses was, unfortunately, not possible because of the already extended duration of the recording sessions (~40 min). However, we measured recovery following 200 Hz stimulation at three different intervals (1, 2 and 4 s) in the ionomycin experiments. Following the suggestion of this reviewer, we now added these experimental data to the graph that illustrates the model predictions. As intended, model predictions are very close to the measured values.

4. Fig. 7D looks as if there is little heterogeneity in the FRP size between individual synapses. I wonder if this result suggests that the FRP size is not a key factor determining F_{TS} ($= M_{TS}/FRP$). It would be helpful to show a plot of FRP size vs. M_{TS} . It should be noted that the RRP size was proportional to the complexity of calyx structure (Fig. 1 of Fekete et al. 2019)

FRP size is not a predictor of the fraction of TS SVs (f_{TS}) as shown in the scatter plots below (**Fig. R4**). Plotting M_{TS} (**Fig. R4A**) or M_{LS} (**Fig. R4B**) as a function of *FRP* reveals strong correlations with a coefficient of determination $r^2 \geq 0.87$. This is trivial as larger presynaptic terminals simply contain more docked SVs (Lin et al., 2011; their Fig. 4A). There is a much weaker correlation between M_{TS} and M_{LS} with $r^2 \sim 0.59$ (**Fig. R4C**). No correlation between f_{TS} and *FRP* was observed (**Fig. R4D**), indicating that LS- or TS-dominated synapses can be found among all synapses regardless of *FRP* size. Because the paper is already quite long, we followed the reviewer's suggestion and added **Fig. R4D** as a new panel C to the current Fig. 2 but omitted the other panels (**Fig. R4A,B,C**).

Figure R4, Linear regression analysis for the data set shown in Figs. 2 and 3 of the manuscript (control data, n=50).

The time courses in **A-E** assume a depolarization of the presynaptic calyx terminal from -80 mV to 0 mV for a duration of 40 ms, starting at $t = 0.4$ s.

A, Scatter plot of M_{TS} versus FRP .

B, Scatter plot of M_{LS} versus FRP .

C, Scatter plot of M_{TS} versus M_{LS} .

D, Scatter plot of f_{TS} versus FRP .

Solid and dotted lines in **A-D** represent linear regressions and 95% confidence limits, respectively.

Minor points:

1. $[Ca^{2+}]$ required for half maximal increase in the TS occupancy is not consistent in this Ms. 438 nM in the text but 617 nM in the legend of figure 6 and Discussion.

Corrected. Thank you.

2. date -> data in page 19

Corrected. Thank you.

3. Page 18, corroborated -> corroborate

Corrected. Thank you.

Referee #2:

In this study, Lin and colleagues explore experimental manipulations that boost synaptic transmission using the framework of a quantitative presynaptic vesicle pool model at the calyx of Held. Specifically, the authors examine the impact of increased calcium influx as well as phorbol esters on synaptic transmission and short-term plasticity using a multi-state kinetic model that distinguishes between two pre-fusion states (LS and TS for loosely docked and tightly docked states, respectively). They conclude that shifts in the balance of TS and LS could account for phorbol-ester induced enhancement of synaptic transmission in addition to the broad diversity of basal synaptic strengths in a population of synapses. In contrast, enhanced calcium influx predominantly increased the probability of fusion with a minor impact on the TS/LS balance. Additionally, phorbol ester treatment increased the SV pool size as estimated using fits to their model. This work is the latest in a series of studies that have reassessed fundamental presynaptic parameters by combining this two-docked-state kinetic model with a computational data fitting method (non-negative tensor factorization). And in agreement with previous papers from this group, the general theme has been to reassign effects traditionally ascribed to vesicle fusion probability to an upstream docking step. Specifically in this study, much of the phorbol ester-induced enhancement of synaptic strength could be accounted for by a shift from the LS to the TS state rather than an increase in SV fusion probability. This interpretation is novel, interesting, and has implications for mechanistic hypotheses underlying phorbol ester enhancement. I have several questions for the authors mainly for clarification purposes.

We were very glad to read Referee #2's overall very positive assessment of our work.

1. One of the early and important points made in this manuscript comes from the observation that the broad variety of synaptic strengths can arise from a population of synapses with essentially identical fusion probabilities but distinct distributions of SVs among the pre-fusion states. If one were to model the synaptic responses to stimulus trains while keeping the TS/LS ratio fixed but allowing for variation in fusion probability, could your kinetic model and NTF approach generate a 'best fit' to the data that is still reasonably good or does it do a far worse job of fitting the data? This may have been done in a previous publication - I was hoping for some objective demonstration that the data is much better described by a fixed p_{Fusion} and variable TS/LS distribution than a fixed TS/LS distribution and a variable p_{Fusion} parameter. Perhaps clarifying this point at the outset will help bolster confidence that there is not an equally valid way of fitting the data with heterogeneous p_{Fusion} accounting for the range of synaptic strengths.

The main purpose of the present study is to validate a previously proposed kinetic scheme for SV priming and fusion by challenging it to reproduce experimentally observed changes in STP.

The rationale for proposing a sequential LS-TS model was already described previously (Neher and Brose, 2018; Neher and Taschenberger, 2021; Lin et al., 2022; Aldahabi et al., 2024; Lopez-Murcia et al., 2024; Neher, 2024). In brief, this kinetic scheme recapitulates the sequential build-up of the synaptic release machinery. It further postulates reversibility for state transitions, which implies that all states are in a dynamic equilibrium. Consequently, release sites are never fully occupied with docked SVs which is in contrast to the frequent assumption of a non-reversible SV priming reaction which implicitly assumes full occupancy of release sites after extended periods of rest. The LS-TS model postulates that only a subpopulation of all docked SVs are in a fully-primed and fusion-competent state, which in turn requires a relatively high p_{fusion} value. The LS-TS model shares this latter property with the RS/DS model recently proposed by the Marty lab (Miki et al., 2018; Silva et al., 2024).

We previously demonstrated, that the profound heterogeneity among calyx synapses in terms of initial synaptic strength and initial depression rate in response to high-frequency stimulation largely disappears when synapses are pre-conditioned with few APs delivered at low f_{stim} (10 Hz). Mechanistically, such finding can be explained either by assuming two parallel pools of fusion-competent SVs with distinctly different fusion probabilities (Taschenberger et al., 2016) or, alternatively, by assuming two sequential pools with quite different resting occupancies while only one pool of SVs is fusion competent (Lin et al., 2022). Unfortunately, an unequivocal differentiation between parallel or sequential kinetic schemes has proven difficult as both types of models often make relatively similar predictions (Weichard et al., 2023).

Importantly, a fixed TS/LS distribution among calyx synapses in combination with a variable p_{fusion} parameter, as proposed by this reviewer, is unable to reproduce the equalizing effect of pre-conditioning on synaptic strength and high-frequency depression.

It is obviously possible to combine a sequential SV priming scheme with the assumption of heterogeneous p_{fusion} among calyx synapses. This increases the model parameter space, and adding more parameters is expected to decrease χ^2 . However, previous work (Lin et al., 2022) and the present manuscript indicate that such additional assumption is not required to faithfully reproduce experimental data.

In sum, we demonstrate that the LS-TS priming scheme, which is mechanistically plausible as it is based on the assumption of a stepwise and reversible build-up of the fusion apparatus, is capable of reproducing various aspects of synaptic transmission and its plasticity which we have studied so far. We cannot and do not claim uniqueness of our model. Instead, we hope that the LS-TS model provides a useful framework for a mechanistic analysis of synaptic plasticity and its modulation at many different types of synaptic connections.

2. One synaptic parameter that has always been difficult to compare across preps and methods is the vesicle pool size. The authors use NTF approaches with their kinetic state model to estimate the fast-releasing SV pool and conclude that phorbol esters increase this pool by around 27%. Numerous past studies have made strong claims about the lack of an effect of phorbol esters on vesicle pool size (e.g. Lou et al 2005 for a calyceal example). Can the authors clarify how they are thinking about these disparities? Is this a matter of methodological differences? Do the authors think that there is a distinct mechanism involved in increasing the FRP compared to the shift from LS to TS? Or are both of these effects likely to derive from the same underlying biochemical changes to Munc13 for instance? And in cases where synapses are assessed using a hypertonic sucrose challenge (other preps and investigators), do the authors believe that both the LS and TS states contribute to the total pool or is this more of a measure of the TS state? Naïve readers may need some help in trying to compare the FRP described here to various distinct methods such as long depolarizing presynaptic steps or sucrose.

This reviewer correctly points out that some earlier studies on calyx synapses found little change in SV pool size upon PDBu treatment. This contrasts the situation at cultured hippocampal synapses for which a SV pool size increase similar to what we describe here was reported (Stevens and Sullivan, 1998; Chang et al., 2010).

As already mentioned by the reviewer, different methods that rely on different assumptions are used to estimate SV pool sizes. Here, we measured the cumulative EPSC in response to high-frequency fiber stimulation (using unperturbed terminals) and estimated the pool size after correction for SV replenishment and incomplete pool depletion. Subsequently, we confirmed the PDBu-induced FRP increase with presynaptic ΔC_m recordings.

Lou et al. (2005) used deconvolution of EPSCs induced by sustained presynaptic depolarizations (using presynaptic whole-cell recordings). They observed an increase in *FRP* size of $11 \pm 8\%$ ($n = 5$; $p = 0.23$). Similar methods were applied Lee et al. (2013), who reported an *FRP* increase of 31% ($n = 8$; $p = 0.002$).

Wu and Wu (2001) used presynaptic ΔC_m measurements to quantify pool size changes in response to PMA treatment. At that time, the concept of a heterogeneous SV pool in calyx terminals (*FRP* + *SRP*) just emerged and Wu and Wu (2001) interpreted their measurements in terms of a single and functionally homogenous SV pool. Their measurements were done with a low Ca^{2+} buffering strength and therefore report the sum of *FRP* + *SRP*. In addition, their measurements may need to be corrected for a possible contributions of SV replenishment to the ΔC_m response. In contrast, we measured ΔC_m responses with 10 mM EGTA in the pipette solution which allowed us to largely abolish release of the *SRP* as well as Ca^{2+} -dependent SV recruitment, thus isolating the *FRP*. Wu and Wu (2001) observed faster pool depletion kinetics in the presence of PMA but little change in the maximum ΔC_m response.

We repeated the experiments of Wu and Wu (2001) using young calyx terminals (**Fig. R5**) and low EGTA (50 μM). Please notice that the total ΔC_m response is about twice as large (*FRP* + *SRP*) as compared to responses obtained with 10 mM EGTA in the pipette (*FRP* only). Consistent with Wu and Wu (2001), **Fig. R5** shows that PDBu increases ΔC_m responses induced by shorter depolarizations more strongly than those induced by longer depolarizations. However, instead of interpreting this finding as a faster depletion kinetics of a homogenous SV pool, we propose that this reflects an increase in the number of fast releasing (*FRP*) SVs, possibly at the expense of slowly releasing (*SRP*) SVs.

Figure R5, Augmentation of release by PDBu in young (P8–10) calyx synapses and with low presynaptic Ca^{2+} buffer.

All recordings with 50 μM EGTA in the pipette solution. These conditions are not suitable to isolate the FRP. Instead, the total ΔC_m responses represent contributions of fast and slowly releasing SVs as well as as newly recruited SVs.

A, Average ΔC_m responses (**A1**) in response to 6 and 24 ms step depolarizations and corresponding average $I_{\text{Ca}(V)}$ (**A2**; 6 ms step) recorded in P8–10 calyx terminals in the absence (*black*) and presence (*red*) of 1 μM PDBu.

B, Pool depletion kinetics in the absence (*black*) and presence (*red*) of 1 μM PDBu. Note the biphasic pool depletion kinetics which reflect contributions of slowly releasing SVs and/or newly recruited SVs.

Release induced by 6 ms steps was more strongly augmented by PDBu than release induced by 24 ms steps (compare *red* and *black* traces in **A1** and **B**). This may be interpreted as an acceleration depletion kinetics of a functionally homogeneous SV pool or, according to our model, as an increase in the fraction of fast releasing SVs.

3. The claim that 25% of the release sites are empty in the basal state seems important but nontrivial to demonstrate. And Figure 4 could use a bit more labeling to help readers navigate the data. How critical is this value of 25% to the authors modeling and conclusions? If that number was 10% or 0%, would the resulting fits be far worse or is this a relatively minor perturbation? I think the manuscript could use some clarification to better describe both the significance of the ES magnitude and the experimental evidence supporting the proposed value.

Varying the fraction of empty sites (N_{ES}/N_{total}) at rest in a certain range leads to similar model predictions, if k_1 is appropriately scaled to keep the product of $k_1 \cdot N_{ES}$ constant. Provided that this is case, predicted release time courses are quite similar for N_{ES}/N_{total} in the range of 15% to 30%. We added more explanation to the Methods section under *Kinetic scheme for SV priming and fusion*.

A fraction of 25% empty sites in resting calyx synapses under control conditions, effectively allows an increase in the total number of TS SVs at rest by two mechanisms: Just increasing the forward rate of the LS→TS transition at rest ($k_{2,rest}$) alone will increase the TS/LS ratio at rest. Increasing both, $k_{1,rest}$ and $k_{2,rest}$, will increase the number of LS SVs and the number of TS SVs at rest, at the expense of empty sites.

In contrast, if the fraction of empty sites is close to zero, any increase in TS SVs at rest has to come solely from an increased $k_{2,rest}$ and will, thus, require very high values for $k_{2,rest}$.

4. What exactly is the TSL state and why is it necessary here? The logic for this particular bit of complexity was not at all clear and left me wondering if you could equally well make all the same points without the TSL state. It appears that a 50% change to the value of a parameter associated with the TSL state was required to fit the ionomycin data, but no alterations of parameters associated with the TSL state were necessary when external Ca or PDBu treatment were modeled.

As described in earlier publications (Lin et al., 2022; Aldahabi et al., 2024; Lopez-Murcia et al., 2024), the TSL represents docking sites occupied with fusion-competent SVs. In Lin et al. (2022) and also in Neher (2024), we pointed out that the TSL had to be postulated in order to achieve net facilitation of release at $f_{stim} = 200$ Hz, i.e. EPSC paired-pulse ratios (PPRs) >1 . The Ca^{2+} -dependent acceleration of the LS→TS transition alone is not sufficient to achieve paired-pulse ratios (PPRs) >1 , if one limits p_{fusion} increases to those returned by NTF analysis. Furthermore, in order to model the fast decay of PPR one had to assume a boost of recruitment, which is only short-lived. The finding by EM of a short-lived state of tight docking or ‘transient SV docking’ (Kusick et al., 2020; Kusick et al., 2022) was exactly what was required to faithfully reproduce release time courses for high

stimulation frequencies. We added more explanation to the Methods section under *Kinetic scheme for SV priming and fusion*.

In the model, every AP triggers the transition of a small fraction (~10%) of LS to TSL. On a ‘free-energy landscape’ (Witkowska et al., 2021; their Fig. 4h), TSL SVs may be located close to the peak on the hill separating LS from TS SVs. Thus, TSL SVs are not stable but rapidly transition back to the (stable) LS state with a time constant of ~90 ms. This rapid back transition implies that the contribution of TSL SVs to release triggered by low-frequency trains is small as the TSL state is effectively emptied during ISIs. As detailed above in the context of **Fig. R3**, the contribution of TSL SVs to release during 200 Hz stimulation is relatively small (about 20% of that contributed by TS SVs).

A main difference between the experiments with ionomycin treatment and those with high external $[Ca^{2+}]_o$ or PDBu treatment is the sustained increase in presynaptic resting $[Ca^{2+}]_i$ following ionomycin application. The observation that ionomycin treatment required a change in κ suggests that the LS→TSL transition may be sensitive to resting $[Ca^{2+}]_i$. Deciphering the molecularly identity of the TSL state needs to be addressed in a future study.

5. Minor comment on Figures 3 and 4 - provide more labeling. For instance, in Fig 3A, perhaps you could eliminate the DeltaCa trace (which isn't adding much) and provide a color legend for stim frequency if that is what the different colors stand for. This shows up in several later figures but may be sufficient just to show a legend once. Some labeling in Figure 4A could help as well.

We added additional labeling as proposed by the reviewer. Thank you for the suggestion.

References

- Aldahabi M, Neher E, Nusser Z (2024) Different states of synaptic vesicle priming explain target cell type-dependent differences in neurotransmitter release. *Proc Natl Acad Sci U S A* 121:e2322550121.
- Bauer PJ (2001) The local Ca concentration profile in the vicinity of a Ca channel. *Cell Biochem Biophys* 35:49-61.
- Chang CY, Jiang X, Moulder KL, Mennerick S (2010) Rapid activation of dormant presynaptic terminals by phorbol esters. *J Neurosci* 30:10048-10060.
- Eisner D, Neher E, Taschenberger H, Smith G (2023) Physiology of intracellular calcium buffering. *Physiol Rev* 103:2767-2845.
- Fedchyshyn MJ, Wang LY (2005) Developmental transformation of the release modality at the calyx of held synapse. *J Neurosci* 25:4131-4140.
- Keller D, Babai N, Kochubey O, Han Y, Markram H, Schurmann F, Schneggenburger R (2015) An Exclusion Zone for Ca^{2+} Channels around Docked Vesicles Explains Release Control by Multiple Channels at a CNS Synapse. *PLoS Comput Biol* 11:e1004253.
- Kochubey O, Han Y, Schneggenburger R (2009) Developmental regulation of the intracellular Ca^{2+} sensitivity of vesicle fusion and Ca^{2+} -secretion coupling at the rat calyx of Held. *J Physiol* 587:3009-3023.
- Kusick GF, Ogunmowo TH, Watanabe S (2022) Transient docking of synaptic vesicles: Implications and mechanisms. *Curr Opin Neurobiol* 74:102535.
- Kusick GF, Chin M, Raychaudhuri S, Lippmann K, Adula KP, Hujber EJ, Vu T, Davis MW, Jorgensen EM, Watanabe S (2020) Synaptic vesicles transiently dock to refill release sites. *Nat Neurosci* 23:1329-1338.

- Lee JS, Ho WK, Lee SH (2012) Actin-dependent rapid recruitment of reluctant synaptic vesicles into a fast-releasing vesicle pool. *Proc Natl Acad Sci U S A* 109:E765-774.
- Lee JS, Ho WK, Neher E, Lee SH (2013) Superpriming of synaptic vesicles after their recruitment to the readily releasable pool. *Proc Natl Acad Sci U S A* 110:15079-15084.
- Lin KH, Oleskevich S, Taschenberger H (2011) Presynaptic Ca^{2+} influx and vesicle exocytosis at the mouse endbulb of Held: a comparison of two auditory nerve terminals. *J Physiol* 589:4301-4320.
- Lin KH, Erazo-Fischer E, Taschenberger H (2012) Similar intracellular Ca^{2+} requirements for inactivation and facilitation of voltage-gated Ca^{2+} channels in a glutamatergic mammalian nerve terminal. *J Neurosci* 32:1261-1272.
- Lin KH, Taschenberger H, Neher E (2022) A sequential two-step priming scheme reproduces diversity in synaptic strength and short-term plasticity. *Proc Natl Acad Sci U S A* 119:e2207987119.
- Lopez-Murcia FJ, Lin KH, Berns MMM, Ranjan M, Lipstein N, Neher E, Brose N, Reim K, Taschenberger H (2024) Complexin has a dual synaptic function as checkpoint protein in vesicle priming and as a promoter of vesicle fusion. *Proc Natl Acad Sci U S A* 121:e2320505121.
- Lou X, Scheuss V, Schneggenburger R (2005) Allosteric modulation of the presynaptic Ca^{2+} sensor for vesicle fusion. *Nature* 435:497-501.
- Miki T, Nakamura Y, Malagon G, Neher E, Marty A (2018) Two-component latency distributions indicate two-step vesicular release at simple glutamatergic synapses. *Nat Commun* 9:3943.
- Müller M, Felmy F, Schwaller B, Schneggenburger R (2007) Parvalbumin is a mobile presynaptic Ca^{2+} buffer in the calyx of held that accelerates the decay of Ca^{2+} and short-term facilitation. *J Neurosci* 27:2261-2271.
- Nägerl UV, Novo D, Mody I, Vergara JL (2000) Binding kinetics of calbindin-D(28k) determined by flash photolysis of caged Ca^{2+} . *Biophys J* 79:3009-3018.
- Naraghi M, Neher E (1997) Linearized buffered Ca^{2+} diffusion in microdomains and its implications for calculation of $[Ca^{2+}]$ at the mouth of a calcium channel. *J Neurosci* 17:6961-6973.
- Neher E (1998) Usefulness and limitations of linear approximations to the understanding of Ca^{++} signals. *Cell Calcium* 24:345-357.
- Neher E (2024) Interpretation of presynaptic phenotypes of synaptic plasticity in terms of a two-step priming process. *J Gen Physiol* 156.
- Neher E, Brose N (2018) Dynamically Primed Synaptic Vesicle States: Key to Understand Synaptic Short-Term Plasticity. *Neuron* 100:1283-1291.
- Neher E, Taschenberger H (2021) Non-negative Matrix Factorization as a Tool to Distinguish Between Synaptic Vesicles in Different Functional States. *Neuroscience* 458:182-202.
- Sakaba T (2006) Roles of the fast-releasing and the slowly releasing vesicles in synaptic transmission at the calyx of held. *J Neurosci* 26:5863-5871.
- Sakaba T, Neher E (2001) Calmodulin mediates rapid recruitment of fast-releasing synaptic vesicles at a calyx-type synapse. *Neuron* 32:1119-1131.
- Silva M, Tran V, Marty A (2024) A maximum of two readily releasable vesicles per docking site at a cerebellar single active zone synapse. *Elife* 12.
- Stevens CF, Sullivan JM (1998) Regulation of the readily releasable vesicle pool by protein kinase C. *Neuron* 21:885-893.
- Taschenberger H, Woehler A, Neher E (2016) Superpriming of synaptic vesicles as a common basis for intersynapse variability and modulation of synaptic strength. *Proc Natl Acad Sci U S A* 113:E4548-4557.

- Wang LY, Neher E, Taschenberger H (2008) Synaptic vesicles in mature calyx of Held synapses sense higher nanodomain calcium concentrations during action potential-evoked glutamate release. *J Neurosci* 28:14450-14458.
- Weichard I, Taschenberger H, Gsell F, Bornschein G, Ritzau-Jost A, Schmidt H, Kittel RJ, Eilers J, Neher E, Hallermann S, Nerlich J (2023) Fully-primed slowly-recovering vesicles mediate presynaptic LTP at neocortical neurons. *Proc Natl Acad Sci U S A* 120:e2305460120.
- Witkowska A, Heinz LP, Grubmuller H, Jahn R (2021) Tight docking of membranes before fusion represents a metastable state with unique properties. *Nat Commun* 12:3606.
- Wu XS, Wu LG (2001) Protein kinase C increases the apparent affinity of the release machinery to Ca^{2+} by enhancing the release machinery downstream of the Ca^{2+} sensor. *J Neurosci* 21:7928-7936.
- Yang CH, Ho WK, Lee SH (2021) Postnatal maturation of glutamate clearance and release kinetics at the rat and mouse calyx of Held synapses. *Synapse* 75:e22215.

Dear Dr Taschenberger,

Re: JP-RP-2025-286282R1 "Number and relative abundance of synaptic vesicles in functionally distinct priming states determine synaptic strength and short-term plasticity" by Kun-Han Lin, Mrinalini Ranjan, Noa Lipstein, Nils Brose, Erwin Neher, and Holger Taschenberger

We are pleased to tell you that your paper has been accepted for publication in The Journal of Physiology.

Yours sincerely,

Katalin Toth
Senior Editor
The Journal of Physiology

If you would like to receive our 'Research Roundup', a monthly newsletter highlighting the cutting-edge research published in The Physiological Society's family of journals (The Journal of Physiology, Experimental Physiology, Physiological Reports, The Journal of Nutritional Physiology and The Journal of Precision Medicine: Health and Disease), please click this link, fill in your name and email address and select 'Research Roundup':
<https://www.physoc.org/journals-and-media/membernews>

- You can help your research get the attention it deserves! Check out Wiley's free Promotion Guide for best-practice recommendations for promoting your work at: www.wileyauthors.com/eoo/guide. You can learn more about Wiley Editing Services which offers professional video, design, and writing services to create shareable video abstracts, infographics, conference posters, lay summaries, and research news stories for your research at: www.wileyauthors.com/eoo/promotion.

EDITOR COMMENTS

Reviewing Editor:

The authors have done an excellent job of addressing the previous critiques. There are no further concerns.

REFEREE COMMENTS

Referee #1:

Authors thoroughly addressed my questions and followed my comments.

I have no further question or comment.

Referee #2:

The authors have addressed my concerns and I support moving forward with publication.